# POINT-BASED INSTANCE COMPLETION WITH SCENE CONSTRAINTS

**Wesley Khademi & Li Fuxin**
Oregon State University
{khademiw, fuxin.li}@oregonstate.edu

## ABSTRACT

Recent point-based object completion methods have demonstrated the ability to accurately recover the missing geometry of partially observed objects. However, these approaches are not well-suited for completing objects within a scene, as they do not consider known scene constraints (e.g., other observed surfaces) in their completions and further expect the partial input to be in a canonical coordinate system, which does not hold for objects within scenes. While instance scene completion methods have been proposed for completing objects within a scene, they lag behind point-based object completion methods in terms of object completion quality and still do not consider known scene constraints during completion. To overcome these limitations, we propose a point cloud-based instance completion model that can robustly complete objects at arbitrary scales and pose in the scene. To enable reasoning at the scene level, we introduce a sparse set of scene constraints represented as point clouds and integrate them into our completion model via a cross-attention mechanism. To evaluate the instance scene completion task on indoor scenes, we further build a new dataset called ScanWCF, which contains labeled partial scans as well as aligned ground truth scene completions that are watertight and collision-free. Through several experiments, we demonstrate that our method achieves improved fidelity to partial scans, higher completion quality, and greater plausibility over existing state-of-the-art methods.

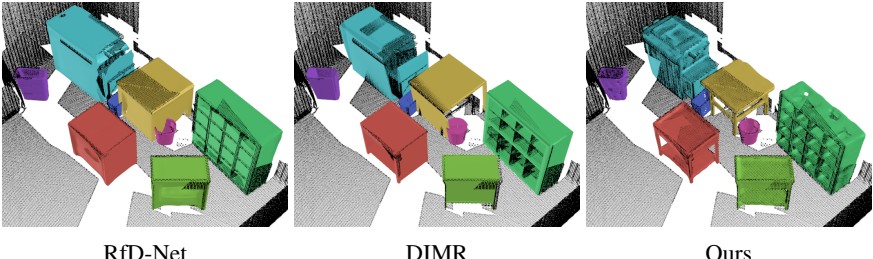

| RfD-Net | DIMR | Ours |

Figure 1: Visual comparison of completion results. Our approach is better at recovering missing geometry, avoiding collisions, and preserving observed surfaces and known free space.

## 1 INTRODUCTION

Object modeling from sensor observations is an increasingly prominent problem for robot interaction within real environments. However, sensors such as LiDAR and depth cameras provide us with only partial observations of objects in a scene. Even with dense scanning and reconstruction, many objects in a reconstructed scene are left incomplete, as parts of their geometry were completely occluded during the scanning process. With a lack of knowledge about the full 3D geometry of objects, the performance of a robot on downstream tasks such as navigation and grasp planning may be hindered because of limited capabilities to infer the center of gravity and other physical properties of objects.

Point-based 3D object completion has been researched for several years starting from Yuan et al. (2018). In recent years, the performance on this task has improved significantly by leverag-

ing encoder-decoder architectures, which first produce a coarse completion, commonly known as *seeds* (Xiang et al., 2021; Zhou et al., 2022) and then upsample it hierarchically. However, most of these work operate in canonically aligned coordinate systems – i.e. setting the center of the *complete object* as having 0 coordinates and a scale of 1, and rotated so that a designated "front" side will always be along the $z$ direction. This is also applied to the partial inputs so that they match the normalization and pose of the completed object. These are unrealistic assumptions when objects are presented in scenes, but they indeed simplify the problem so that, in some sense, algorithms only have to match the partial object to complete objects in the training set, then directly generate the coordinates for the completion, as the canonical coordinates of the complete object will always be the same no matter which part is used as input.

On the other hand, semantic scene completion (Song et al., 2017; Liu et al., 2018; Li et al., 2019; Dong et al., 2023) has been studied heavily, but they only provide voxel labels without identifying objects. Hou et al. (2020) proposed the instance scene completion task, which involves detecting individual object instances in the scene and completing the missing regions of these objects. There has been some work on this front (Nie et al., 2021; Tang et al., 2022; Li et al., 2023a), but the network structures have not been explored as much as the point-based object completion task.

In this paper, we explore whether the more refined network structures of point-based object completion models can be adapted to the instance scene completion task. We note two hurdles to this goal, the first is to lift the canonical coordinates assumption so that the completion model no longer has to know the scale and pose of the full object. The second is to make the instance completion to be aware of potential scene constraints that can be deduced from visibility. For example, if one sees a surface, then the network should not add points on this viewing ray. Besides, if there is another object already present at some location, then the completion should not collide with those areas.

To solve these two difficult problems, we significantly improve the point-based completion framework mainly by adopting a more sophisticated seed generator. First, we change the seed prediction to two parts, predicting an object center location and seed offsets from the center. We made several architectural improvements to improve performance in this setup to match the completion performance when using canonical coordinates. Next, to provide our completion model with scene context, we introduce a set of sparse constraints which encode known information about the scene. Unlike dense TSDFs, our constraints are represented as two bounding shells of the underlying surface, indicating the transition boundary from surface to known free space and surface to known occluded space. We integrate these constraints into our seed generator, enabling our model to reason about completions which are plausible in the context of the observed scene. Using this additional scene information, we find that our approach produces fewer collisions between predicted completions.

Furthermore, existing datasets for the instance scene completion task suffer from errors in the ground truth data, making evaluation on them unreliable. Scan2CAD (Avetisyan et al., 2019a) lacks alignment between real partial scans from ScanNet and synthetic ground truth meshes from ShapeNet, causing a trade-off between respecting the partial scan and matching the ground truth mesh. Scan-ARCW (Li et al., 2023a) contains collisions in their ground truth data, making it difficult to measure the plausibility of scene completion with collision metrics. We introduce a new dataset called Scan-WCF addressing the issues present in these existing datasets. On our newly proposed dataset we demonstrate that our approach outperforms existing works in terms of both partial reconstruction quality and completion quality while producing less collisions between predictions.

In summary, our main contributions are as follows:

- We propose a novel object-level completion model which is robust to the scale and pose of objects found within scenes.
- We integrate a sparse set of scene constraints into our model to provide our object completions with scene context (e.g., other observed surfaces, free space, occluded space).
- We build a new dataset for instance scene completion on indoor scenes, containing partial scans and watertight ground truth meshes that are aligned, labeled, and collision free.

## 2 RELATED WORK

The 3D completion literature can be roughly broken down into two main categories: (1) object-level completion and (2) scene-level completion.

**Object-level Completion** 3D object completion aims at recovering the complete geometry of an object given some partial observation. With the introduction of point cloud architectures, Yuan et al. (2018) was the first to propose a fully point cloud based completion architecture. Since then, there have been many follow-up works which leveraged increasingly better encoder-decoder architectures (Tchapmi et al., 2019; Liu et al., 2020; Wen et al., 2020; Huang et al., 2020; Wen et al., 2021; Yu et al., 2021). In recent years, a popular choice has been to first produce a coarse completion, often referred to as seeds, and then upsample the completion in a hierarchical fashion (Xiang et al., 2021; Zhou et al., 2022; Chen et al., 2023; Khademi & Fuxin, 2024). However, all these approaches operate in canonically aligned coordinate systems and thus are not suited for completing objects in the context of scenes where pose and scale are arbitrary. SCARP (Sen et al., 2023) is a point cloud completion method that aims to be robust to arbitrary pose by predicting the completion in a canonical frame along with the pose needed to transform the completion back to its true coordinate system. Unlike SCARP, our method considers the scene context around the object as we are interested in the scene completion task. Additionally, our approach does not require estimating the pose of the object, avoiding possible misalignment with the partial scan due to inaccuracies in pose estimation.

**Scene-level Completion** Scene-level completion has heavily been studied as a joint task of semantically labeling the scene while recovering the geometric structures that are missing from it. For indoor environments, this task has been studied almost exclusively as predicting the semantic label and occupancy of each voxel in a dense voxel grid (Song et al., 2017; Liu et al., 2018; Li et al., 2019; Chen et al., 2020; Cai et al., 2021; Dong et al., 2023; Wang et al., 2024). These methods do not assign voxels to object instances in their completion, and use dense 3D convolutions to produce predictions on low-resolution voxel grids, prohibiting the representation of fine-grained geometry typically present in indoor environments.

Avoiding the need for dense occupancy and semantic prediction, some methods produce completions of partial scans by performing CAD model retrieval and alignment (Avetisyan et al., 2019b; Ishimtsev et al., 2020). However, these approaches require the existence of a CAD pool, have to search for the nearest CAD model for each object in the partial scan, and potentially require an additional optimization for roughly aligning the models to the scan.

To improve the level of scene understanding required for object interaction within environments, Hou et al. (2020) proposed the instance scene completion task, which involves detecting individual object instances in the scene and completing the missing regions of these objects. Their approach relies on producing occupancy predictions and semantic labels for a dense voxel grid using 3D convolutions, limiting the resolution of scenes due to memory footprint. More recent instance scene completion methods operate directly on point clouds. RfD-Net (Nie et al., 2021) generates instance proposals via 3D object detection and completions via an implicit function. DIMR (Tang et al., 2022) trains a 3D instance segmentation model for proposal generation and produces completions from latent codes with a pre-trained shape generator. DDIT (Li et al., 2023a) performs 3D instance segmentation and deforms deep implicit shape templates into completions conditioned on shape latent codes extracted from the segmented partial objects. However, DDIT requires an iterative procedure for estimating object pose and a per-scene optimization step for respecting the partial input which is slow. PaSCo (Cao et al., 2024) performs panoptic scene completion on LiDAR scans of outdoor scenes. However, they employ a dense 3D CNN in part of their network, making their work infeasible for our indoor scenes due to memory constraints (our 2cm resolution indoor scans would require $\sim 4 - 5\times$ more voxels than the outdoor scans with 20cm voxels used in their work).

## 3 METHOD

We present an overview of our architecture in Figure 2. Given a partial scan of a scene, we run a state-of-the-art 3D instance segmentation method Mask3D (Schult et al., 2023) to decompose our scene into a set of objects. The partial information of each object is first encoded into a set of local features and a global shape descriptor. We then introduce a novel seed generator which generates a coarse representation of the complete shape, called Patch Seeds, from our partial encoding. Our generator produces Patch Seeds as offsets of the object's predicted center, which we find to be more robust to change in pose compared to existing Patch Seed generators. To provide our object completions with scene context, we additionally integrate a set of scene constraints into our seed generator via cross-attention. Generated Patch Seeds are then decoded into a dense completion in a

Figure 2: Overview of our instance scene completion framework. Instance segmentation is first performed on the partial scan to decompose the scene into its individual objects. Each object is run through our proposed object completion model, which predicts both the complete shape and surface normals. Meshes of each object are then reconstructed to produce the completed scene.

hierarchical fashion by applying a series of upsampling layers. We design our upsampling layer to contain both local and global attention, which helps with producing a globally coherent completion capable of representing fine-grained geometry. At the densest completion level, we additionally leverage a local transformer to predict surface normals, allowing us to reconstruct meshes of our completions using an off-the-shelf surface reconstruction method. Finally, we place each mesh back into the world frame, producing a scene of labeled object instances with complete geometry. In the following sections, we describe the key components of our completion model in more detail.

## 3.1 PARTIAL ENCODER

Our partial encoder takes as input the partial object instance $\boldsymbol{P} \in \mathbb{R}^{M \times 3}$ and estimated surface normals $\boldsymbol{N} \in \mathbb{R}^{M \times 3}$ to produce a downsampled set of points $\boldsymbol{P}^l \in \mathbb{R}^{M_l \times 3}$ with local features $\boldsymbol{F}_p^l \in \mathbb{R}^{M_l \times C_{local}}$ and a global shape descriptor $\boldsymbol{f}_p \in \mathbb{R}^{C_{global}}$ from the object. We base our encoder on the design proposed by Khademi & Fuxin (2024), which is in turn based on Zhou et al. (2022). The encoder consists of $l$ downsampling blocks, where at each block the point set is first downsampled, followed by a series of point convolutions for interpolating and aggregating features for the downsampled point set. The final downsampled set of features are then passed through an MLP followed by a max-pooling operation to produce global shape descriptor $\boldsymbol{f}_p$.

The main improvement we made to the partial encoder is that we replace the PointConv (Wu et al., 2019) layers in Khademi & Fuxin (2024) with VI-PointConv (Li et al., 2023b) layers and additionally use estimated normals as input. The convolution filters generated in PointConv are only translation invariant. To be robust to rotation and scale, the network would have to be shown the same object under many different rotations and scales and have enough capacity to encode appropriate filter weights for each of them. On the other hand, the convolution filters in VI-PointConv are generated from a mix of non-invariant, scale-invariant, and rotation-invariant position embeddings. In this way, the MLP generating filter weights can potentially learn to ignore the non-invariant position embeddings to share filter weights across neighborhoods of different scales and rotations, increasing robustness of the network. Furthermore, using both XYZ coordinates and surface normals provide a better description of local surface geometry than XYZ coordinates alone.

## 3.2 SEED GENERATOR

Given the downsampled point set $\boldsymbol{P}^l$, locally extracted features $\boldsymbol{F}_p^l$, and global shape descriptor $\boldsymbol{f}_p$, our seed generator is tasked with producing a set of Patch Seed coordinates $\boldsymbol{S} \in \mathbb{R}^{M_{seed} \times 3}$ and features $\boldsymbol{F}_{seed} \in \mathbb{R}^{M_{seed} \times C_{seed}}$ which represent a coarse encoding of the complete shape.

SeedFormer (Zhou et al., 2022) produces Patch Seeds through a local attention-based upsampler followed by an MLP to regress the seed coordinates. In Table 3 we show that such an approach suffers a significant drop in completion quality if we attempt to use it to complete objects not canonically aligned along an axis. Instead, we design a global attention-based seed generator, which allows the generator to use information from the entire scene to predict the center of the object before producing seed coordinates as offsets of the object center. Our full architecture is shown in Figure 3.

We introduce a learnable token $\boldsymbol{o}_{token} \in \mathbb{R}^C$ which is to be decoded into the center of an object $\boldsymbol{O} \in \mathbb{R}^3$. Our learnable token $\boldsymbol{o}_{token}$ along with downsampled partial information $\{\boldsymbol{P}^l, \boldsymbol{F}_p^l\}$ is first passed through a set of transformer blocks performing multi-head self-attention to produce an updated object center embedding $\boldsymbol{o}_{obj} \in \mathbb{R}^C$ and updated partial features $\boldsymbol{F}_{obj} \in \mathbb{R}^{M_l \times C}$. Having

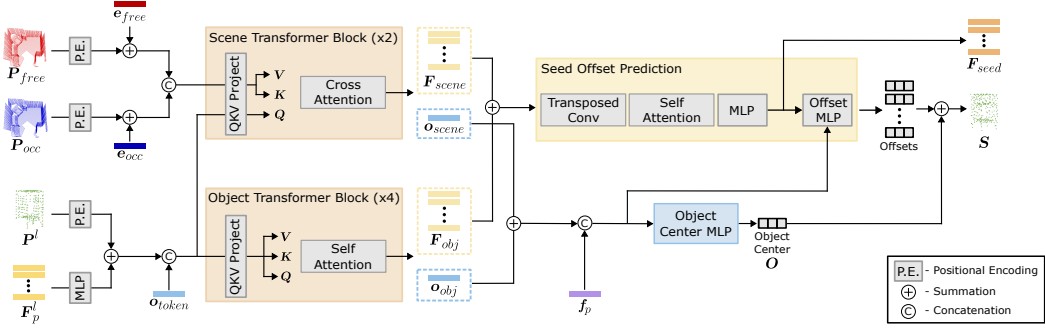

Figure 3: Overview of our proposed seed generator. Predicting Patch Seed coordinates as offsets from the shape's predicted object center is more robust than directly regressing seed coordinates as in Zhou et al. (2022). Our object completions additionally consider other objects in the scene through cross-attention with our known free and occluded space constraints.

aggregated information from the partial input via attention, $\boldsymbol{o}_{obj}$ is then used to regress the object center. We concatenate $\boldsymbol{o}_{obj}$ with the global shape descriptor $\boldsymbol{f}_p$ along the feature dimension before predicting the object center $\boldsymbol{O}$ with an MLP $\theta$:

$$\boldsymbol{O} = \theta([\boldsymbol{o}_{obj}, \boldsymbol{f}_p]) \tag{1}$$

To reliably cover the entire object in our coarse representation, we increase the number of points present in our Patch Seeds compared to the partial points and features from which they are generated. Specifically, we "split" our output partial features $\boldsymbol{F}_{obj}$ using a transposed convolution with a stride and kernel size of 2. Upsampled features are then passed through a multi-head self-attention block followed by an MLP $\omega$ to produce Patch Seed features $\boldsymbol{F}_{seed}$:

$$\boldsymbol{F}_{seed} = \omega(\text{SelfAttn}(\text{TransposedConv}(\boldsymbol{F}_{obj}))) \tag{2}$$

Finally, Patch Seed coordinates $\boldsymbol{S}$ are predicted as offsets from object center $\boldsymbol{O}$. We use a MLP $\gamma$ which predicts per point offsets from Patch Seed features $\boldsymbol{F}_{seed}$ concatenated with object center token $\boldsymbol{o}_{obj}$ and global shape descriptor $\boldsymbol{f}_p$ along the feature dimension:

$$\boldsymbol{S} = \boldsymbol{O} + \gamma([\boldsymbol{F}_{seed}, \boldsymbol{o}_{obj}, \boldsymbol{f}_p]) \tag{3}$$

## 3.3 SCENE-AWARE OBJECT COMPLETION

As mentioned earlier, correctly completing objects in a scene requires satisfying visibility constraints from the scene, i.e. not creating object parts that cut into other objects or grow towards the camera (violating the free space that has been seen). TSDFs are commonly used for representing known information in a scene, such as the surfaces of objects as well as known free or occluded space due to the viewpoints at which the scene was captured. However, TSDFs are typically stored as dense voxel grids, making them computationally expensive to process and not compatible with point clouds. To our knowledge, no one has attempted to address the difficult task of incorporating scene constraints to point-based completion. We propose to represent the constraint information in the scene as sparse sets of points representing known occupied and free spaces. Our goal is to input these constraint points to the network so that it can learn to avoid generating parts in those areas.

We generate our scene constraints as two bounding shells of the surface of the partial scan defined as $\boldsymbol{P}_{in} \pm \delta \boldsymbol{N}_{in}$, where $\boldsymbol{P}_{in} \in \mathbb{R}^{M_{in} \times 3}$ is the partial scan with surface normals $\boldsymbol{N}_{in} \in \mathbb{R}^{M_{in} \times 3}$ and $\delta \in \mathbb{R}$. We further resample these surfaces to 10 cm resolution, leaving us with a sparse set of free space points $\boldsymbol{P}_{free} \in \mathbb{R}^{M_{free} \times 3}$ and occluded space points $\boldsymbol{P}_{occ} \in \mathbb{R}^{M_{occ} \times 3}$. Along with these point sets, we learn additional embeddings $\boldsymbol{e}_{free} \in \mathbb{R}^C$ and $\boldsymbol{e}_{occ} \in \mathbb{R}^C$, which are shared across all free space points and occluded space points, respectively.

We provide our constraints as information to our seed generator through a set of transformer blocks containing multi-head cross-attention as shown in Figure 3. Here the object's partial information is treated as the query tokens while the scene constraints represent the key-value pairs, allowing each partial object to decide what constraints to focus on when completing the object. The transformer blocks output a set of features $\boldsymbol{F}_{scene} \in \mathbb{R}^{M_l \times C}$ and $\boldsymbol{o}_{scene} \in \mathbb{R}^C$, which are directly added to the outputs $\boldsymbol{F}_{obj}$ and $\boldsymbol{o}_{obj}$, respectively, before being used to predict the object's center and Patch Seeds.

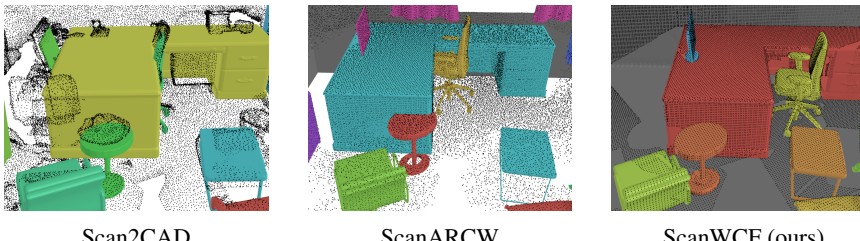

Scan2CAD      ScanARCW      ScanWCF (ours)

Figure 4: Our proposed ScanWCF has aligned ground truth meshes and partials scans while being free of collisions, unlike previous datasets Scan2CAD and ScanARCW.

### 3.4 COARSE-TO-FINE DECODER

For upsampling, we use the upsampling layer proposed by Zhou et al. (2022). In each upsampling layer, we further introduce a series of global attention layers before their local attention-based Up-sample Transformer. With our added global attention, we provide the refinement and upsampling layers with information which can encourage global coherence across the completion. Additionally, global attention allows us to potentially learn fine-grained structure for the missing geometry from the existing geometry which may not be present in the local neighborhood (e.g., through a symmetric part on the opposite side of the object).

Starting from our Patch Seed coordinates $S$ and features $F_{seed}$, we repeatedly apply our upsampling layer to generate a dense completion in a coarse-to-fine manner. At each upsampling layer $j$, we produce a completion $C^j$ and the corresponding upsampled features $F_{up}^j$ with double the resolution of the previous layer. We apply 3 upsampling layers to upsample our completion from a coarse resolution of $M_{coarse} = 256$ points to $M_{dense} = 2,048$ points. The completion produced at each layer is supervised by the ground truth completion $C_{gt}^j$ which has been subsampled to the same resolution as the output resolution at layer $j$ (we additionally treat the Patch Seed coordinates $S$ as our coarsest resolution $C^0$ during training).

### 3.5 MESH RECONSTRUCTION

We used NKSR (Huang et al., 2023) to reconstruct meshes from point clouds. NKSR requires both point clouds and surface normals, where normals are typically estimated from the point cloud using PCA-based plane fitting. In practice, we found these normals to be overly noisy, leading to poor reconstruction quality on our completions. To address this, we introduce a normal estimation module which is jointly trained with our completion network. We first process our final completion $C^3 \in \mathbb{R}^{M_{dense} \times 3}$ and corresponding upsampled features $F_{up}^3 \in \mathbb{R}^{M_{dense} \times C}$ using a modified version of the Upsample Transformer from SeedFormer (Zhou et al., 2022), replacing the transposed convolution used for upsampling with a regular convolution. The features output from this layer encode local surface information of the object and are directly used as input to a small MLP that regresses surface normals $N^3 \in \mathbb{R}^{M_{dense} \times 3}$.

### 3.6 LOSS

We use the same loss function for both pre-training the object completion model and training our scene completion model. Our overall loss objective is defined as:

$$\mathcal{L} = \lambda_c \sum_{j=0}^{3} \mathcal{L}^{CD}(C^j, C_{gt}^j) + \lambda_p \sum_{j=0}^{3} \mathcal{L}^{OCD}(P, C^j) + \lambda_o \mathcal{L}^{MSE}(O, O_{gt}) + \lambda_n \mathcal{L}^{CS}(N^3, N_{gt}^3) \quad (4)$$

where $\mathcal{L}^{CD}$ is the Chamfer Distance between predicted and ground truth completions at each up-sample layer $j$, $\mathcal{L}^{OCD}$ is the One-sided Chamfer Distance between partial input and completion at each upsample layer $j$, $\mathcal{L}^{MSE}$ is the mean squared error between predicted object center and ground truth object center $O_{gt} \in \mathbb{R}^3$, and $\mathcal{L}^{CS}$ is a cosine similarity based loss between the predicted normals $N^3$ and ground truth normals $N_{gt}^3$. We set $\lambda_c = 1$, $\lambda_p = 1$, $\lambda_o = 1$, $\lambda_n = 10^{-2}$ for experiments.

## 4  SCANWCF DATASET

Previous works evaluate the instance scene completion task on the Scan2CAD dataset (Avetisyan et al., 2019a). The Scan2CAD dataset is derived from both the ScanNet (Dai et al., 2017) and ShapeNet (Chang et al., 2015) datasets, where for each detected partial object in a ScanNet scene, the closest synthetic mesh from ShapeNet has been selected and fit to the scan to serve as the ground truth completion. The synthetic meshes are not the true completion of the partial object and their fit to the partial scan is imperfect as shown in Figure 4. The lack of alignment between the input and ground truth makes the evaluation of metrics unreliable on this data. To address this, Li et al. (2023a) proposed ScanARCW which regenerates new partial scans by rendering depth maps and semantic labels of the ground truth meshes and then backprojects this information back into 3D to generate an aligned scan. While ScanARCW addresses the alignment issue, the ground truth scenes in their dataset contain collisions. This makes it unreliable to measure scene completion plausibility with collision metrics, as the collision may be an artifact of the ground truth data itself.

To address the limitations of existing datasets, we introduce a new dataset called ScanWCF, where "WCF" refers to our ground truth scenes being "Watertight and Collision Free". Our dataset contains scenes with partial scans and ground truth complete meshes that are aligned and labeled, with each ground truth scene being watertight and collision free. We generate our scenes from data included in both the Scan2CAD and ScanARCW datasets. We use the background meshes from ScanARCW as scene boundaries (e.g., walls, floor, ceiling), filling holes in the mesh to make it completely enclosed. For selecting which objects to place in the scene, we use the Scan2CAD object matchings. Each ground truth mesh is processed to be watertight and initialized in the scene using the pose and scale from Scan2CAD. We then optimize the pose and scale of each object in the scene such that they are: (1) closely aligned with the partial scan from ScanNet, (2) do not contain any collisions, and (3) are not floating in air. After optimization, we manually verify the scene is collision-free before including it in our dataset. If there exist minor implausibilities after optimization, we manually correct the scene; otherwise we completely discard it if too many collisions exist. After our optimization and manual verification, we find that on average only $0.14\%$ of the points in the scene are in collision with another object, compared to an average of $2.5\%$ of points per scene in the ScanARCW dataset. Finally, to generate aligned and labeled partial scans, we render depth maps and instance segmentation maps from a subset of camera poses in the ScanNet camera trajectory. This information is then backprojected into 3D to generate a partial scan labeled with instance information.

Our dataset contains 1202 indoor scenes based on ScanNet scenes, where 946 of the scenes are used for training and the other 246 scenes are reserved for testing. To increase the amount of training and test data, we generate 2 partial scans per scene using a different subset of camera poses from the ScanNet camera trajectory. We refer readers to our appendix for more details.

## 5  EXPERIMENTS

In this section, we perform a variety of experiments to demonstrate the superiority of our method over existing approaches. We first evaluate our method on the task of instance scene completion, where the goal is to jointly predict object instances and their completions. Secondly, we evaluate completion quality in isolation by removing the instance prediction task and instead providing the ground truth instance information directly to each method. Across both tasks, we show that our method outperforms existing approaches both quantitatively and qualitatively. Finally, we conduct a set of ablations which validate the design choices of our architecture.

### 5.1  IMPLEMENTATION DETAILS

Our object-level completion model is pre-trained on the 34 ShapeNet categories present in our dataset, excluding objects present only in the validation scenes. For augmentations, we perform random rotations about the up-axis. For optimization, we use Adam with an initial learning rate of $1 \times 10^{-4}$ and linearly decay by a factor of $0.98$ every 2 epochs. We train for 150 epochs with a batch size of 64 using 2 NVIDIA V100 GPUs, which takes approximately 4 days. Our scene completion model is then trained on the ScanWCF dataset, initialized from the weights of our pre-trained object completion model. For training, we use the same augmentation and optimization setup as used for pre-training. We train for 200 epochs on a single RTX 4090 GPU which takes about 3 days.

Table 1: **Instance scene completion quality**. Mean average precision (mAP) is reported for different *metric@threshold*.

|  | IoU@0.25 | IoU@0.5 | CD@0.1 | CD@0.047 | LFD@5000 | LFD@2500 | PCR@0.5 | PCR@0.75 |
|---|---|---|---|---|---|---|---|---|
| RfD-Net | 38.78 | 7.25 | **61.68** | 31.32 | 39.18 | 14.65 | 59.56 | 42.36 |
| DIMR | 34.18 | 8.67 | 41.15 | 23.30 | 27.14 | 8.81 | 43.37 | 31.14 |
| Ours (no pre-training) | 61.99 | 45.13 | 61.48 | 54.67 | **52.03** | 24.57 | **63.71** | **62.69** |
| Ours (w/ pre-training) | **62.77** | **47.55** | 61.57 | **56.07** | 51.29 | **29.03** | **63.71** | 62.67 |

## 5.2 BASELINES

We evaluate our proposed approach against instance scene completion methods RfD-Net (Nie et al., 2021) and DIMR (Tang et al., 2022). Both methods are trained on our ScanWCF dataset using the same hyperparameters that were used for their original experiments on the Scan2CAD dataset. We additionally retrain the pre-trained shape generator used in DIMR, but on the 34 categories from ShapeNet present in our dataset rather than the 8 ShapeNet categories it was originally trained on.

## 5.3 METRICS

For the instance scene completion task, we follow Tang et al. (2022) and compute the mean average precision (mAP) across several different metrics and at varying thresholds. Specifically, Intersection over Union (IoU), Chamfer Distance (CD), and Light Field Distance (LFD) are used for evaluating completion quality while the Point Coverage Ratio (PCR) metric is used for evaluating partial reconstruction quality. More details of those metrics are presented in the appendix. For IoU and PCR, higher thresholds are more challenging, while for CD and LFD lower thresholds are more difficult.

To remove the possibility that poor completions are caused by worse instance segmentation, we evaluate completion quality when each method uses the ground truth instance masks. For partial reconstruction quality, we compute Unidirectional Hausdorff Distance (UHD) and One-Sided Chamfer Distance between the partial scan and the predicted complete scene. To evaluate completion quality, we use the Chamfer Distance (CD) between points uniformly sampled from the scene mesh and the predicted completion. Finally, we evaluate the plausibility of scene completions by measuring collisions between predicted completions. For each object, we sample points from its completion and compute their signed distance to all the other predicted objects in the scene and the scene background mesh, penalizing points which fall inside another object mesh or outside the background mesh. We measure both the average distance of collisions (COL) and the percentage of points in collision (%COL). We scale UHD, One-Sided CD, and CD by $10^3$ and scale COL by $10^4$.

## 5.4 RESULTS

**Instance Scene Completion** We present results on the instance scene completion task in Table 1. Despite the fact that we do not jointly train our instance segmentation network together with our completion model, unlike RfD-Net and DIMR, we outperform both approaches in terms of mAP across almost all metrics and thresholds. Both RfD-Net and DIMR suffer large drop-offs in performance for all metrics in the transition from easier threshold to more difficult threshold. This suggests that these approaches can produce completions that share some similarities to the ground truth shape, but likely cannot represent fine geometric details while also suffering from inaccuracies in their prediction of an object's pose and scale. On the other hand, we see much smaller drop-offs in our method for both CD and PCR, suggesting better ability to represent the shape's surface accurately. Furthermore, the pre-training of our object completion model is not a strict requirement,

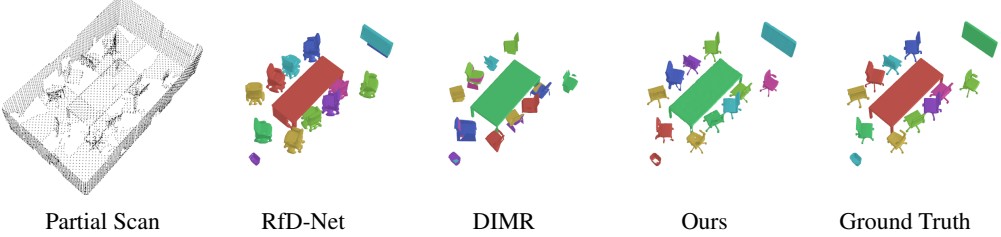

| Partial Scan | RfD-Net | DIMR | Ours | Ground Truth |

Figure 5: Qualitative comparison on the instance scene completion task.

Table 2: **Scene completion quality**. Evaluation of scene completion quality metrics when ground truth instance masks are used for all methods.

|  | UHD ↓ | One-sided CD ↓ | CD ↓ | COL ↓ | % COL ↓ |
|---|---|---|---|---|---|
| RfD-Net | 190.65 | 38.39 | 38.36 | 8.58 | 3.79 |
| DIMR | 270.67 | 44.91 | 39.04 | 10.42 | 4.63 |
| Ours (no pre-training) | 60.56 | 12.50 | 21.22 | 3.42 | **1.74** |
| Ours (w/ pre-training) | **60.07** | **12.10** | **20.74** | **2.67** | 1.81 |

as we outperform both RfD-Net and DIMR without it; however, we do find that pre-training helps improve completion quality at the more challenging metric thresholds.

In Figure 5 we share an example of instance scene completions on a partial scan. RfD-Net struggles to represent thin geometric structures such as the base of a rolling chair, merging the wheel bases into almost a solid circular base. DIMR suffers from low-fidelity completions, with many objects appearing to be a composition of planar primitives and failing to produce thin structures such as the legs of the chairs. On the other hand, our approach can faithfully represent the fine-grained geometry present in the partial scans while producing plausible hallucinations of the missing regions (e.g., wheel bases of chairs). We share more qualitative results in Figures 9 and 10 of our appendix.

**Scene Completion** To isolate our evaluation of completion quality, we present results on scene completion when the ground truth instance information has been provided to each method. In this setting, poor completion quality cannot be attributed to incorrect instance predictions, enabling us to better understand the limitations of the completion network of each method. In Table 2, One-Sided CD and UHD measure the predicted completions average and maximal deviation from the surface of a partial scan, respectively. Both of these metrics indicate that RfD-Net and DIMR are significantly worse at respecting the partial input in comparison to our approach. Similarly, we find that our method is capable of producing higher quality completions than previous approaches, reflected by the significant gap in CD. Finally, our method does a much better job at avoiding collisions between predictions. In particular, RfD-Net and DIMR tend to produce completions that penetrate much further into other objects in the scene, producing an average collision distance (COL) which is $3 - 4\times$ larger than our method. This is further demonstrated by RfD-Net and DIMR having $2 - 3\%$ more points from predicted completions either in collision or extending outside the scene boundaries compared to our approach. In Figures 1 and 6, we qualitatively demonstrate that our approach achieves higher fidelity to the partial scan, better completion quality, and produces less collisions than previous methods. We refer readers to our appendix for more qualitative results.

## 5.5 Ablation Studies

In Table 3, we conduct an ablation of our object completion model on the chair category from ShapeNet to justify our proposed additions for completing objects under arbitrary pose and scale. Our baseline method is SeedFormer (Zhou et al., 2022) with their partial encoder replaced by the PointConv encoder proposed in (Khademi & Fuxin, 2024). This replacement suffers no drop in performance and allows us to later update the PointConv layers for VI-PointConv layers in our work. When trained and tested under ideal conditions (i.e., normalizing the input by the ground truth complete shape and having the object canonically aligned along a shared axis) our baseline is capable of producing high quality completions. When our baseline is trained and evaluated on a more realistic setting (i.e., normalizing by the partial input and objects containing arbitrary pose and scale) we see that performance drops in both completion and partial reconstruction quality. We find that adding VI-PointConv (Li et al., 2023b) to our partial encoder improves partial reconstruction quality, as the added rotation and scale invariant features likely enable better reasoning about the partial input.

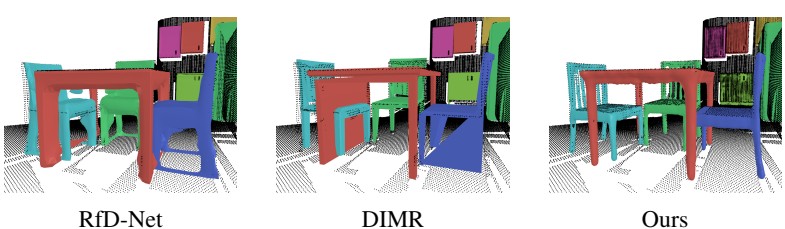

| RfD-Net | DIMR | Ours |
|---|---|---|

Figure 6: Results on scene completion when ground truth instance information is provided.

Table 3: Ablation of our proposed object completion network. Each row containing + additionally contains all the modules in the rows above it.

|  | One-sided CD ↓ | CD ↓ (Seeds) | CD ↓ (Dense) |
|---|---|---|---|
| Baseline using canonical coordinates | 6.50 | 41.69 | 20.20 |
| Baseline | 7.12 | 47.00 | 22.87 |
| + VI-PointConv & Input Normals | 6.04 | 46.47 | 22.57 |
| + Object Center & Seed Offset Prediction | 6.28 | 42.86 | 21.11 |
| + Global Attention in Upsample Layers | 6.18 | 43.71 | 20.05 |
| + Surface Normal Prediction | 6.19 | 43.26 | 20.08 |

Table 4: Ablation of our scene completion network with and without scene constraints.

|  | UHD ↓ | One-sided CD ↓ | CD ↓ | COL ↓ | % COL ↓ |
|---|---|---|---|---|---|
| w/o constraints | 60.18 | 12.20 | 22.37 | 3.75 | 1.83 |
| w/ constraints | 60.07 | 12.10 | 20.74 | 2.67 | 1.81 |

| W/out constraints | W/ constraints | W/out constraints | W/ constraints |

Figure 7: Comparison of our completion model with and without considering scene constraints.

Additionally, with the introduction of global attention in the seed generator and producing seed coordinates by first regressing the object center followed by predicting seeds as offsets from the center, we see the coarse and dense completion quality significantly improve over the old seed generator. Furthermore, incorporating global attention into the upsampling layers allows us to not only match our baseline's performance under ideal conditions, but actually beat it even though we are under non-ideal conditions. Finally, we find that our surface normal prediction module neither harms nor helps our completions; however, it provides us with a way to reconstruct object meshes.

In Table 4, we evaluate the importance of leveraging scene constraints. We observe that using scene constraints does not improve partial reconstruction quality (UHD and One-Sided CD) as the completion model does not need information about free space and occluded space to reconstruct the already observed portion of the object. However, we find that incorporating scene constraints into our model produces a 7% relative improvement in completion quality (CD) and 29% relative improvement in how far points in collision are penetrating into each other (COL). The scene constraints provide our model with information about the scene boundaries as well as other objects in the scene, which helps better constrain our completions to ones which are plausible within the scene as shown in Figure 7.

## 6 CONCLUSION

We present a novel scene completion framework which obtains state-of-the-art performance on the instance scene completion task for indoor scenes. Our proposed object completion model is robust to arbitrary pose and scale, enabling our method to produce high-quality completions with high fidelity to the partial input without having to rely on accurately estimated pose and scale parameters needed to transform the object to a canonically aligned axis. Furthermore, our proposed scene constraints enable our completion model to incorporate scene context in our completions, improving completion quality and reducing collisions between predicted completions. To evaluate our approach, we build a new dataset for the instance scene completion task on indoor scenes called ScanWCF, which contains collision-free scenes whose partial scans and ground truth meshes are aligned and labeled. Through several experiments, we demonstrate our method achieves higher completion quality, greater fidelity to the partial scan, and better plausibility over existing approaches.

Our completion framework is deterministic, meaning that our approach can only produce one completion of a partial scan. In the future, we plan to explore incorporating generative models into our completion framework in order to be able to produce multiple plausible completions of a scene.

ACKNOWLEDGMENTS

This work is partially supported by NSF under grants 1751402, 2321851, ONR under grant N0014-21-1-2052, and DARPA under grant HR00112490423.

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

# A   APPENDIX

An overview of our appendix is presented as follows:

- Model details (Section B): we provide more details of our scene completion model
- ScanWCF dataset (Section C): we provide a detailed description of our proposed dataset
- Metrics (Section D): we formally define our evaluation metrics
- Results (Section E): we share more results and ablations of our method

# B   MODEL DETAILS

## B.1   PARTIAL ENCODER

Our partial encoder consists of $l = 4$ downsampling blocks, where each block contains a downsampling operation on the point set followed by 2 VI-PointConv (Li et al., 2023b) layers. For point convolutions, we use a neighborhood size of 16. We design our encoder such that the final downsampled point set $\boldsymbol{P}^4 \in \mathbb{R}^{M_4 \times 3}$ and corresponding local features $\boldsymbol{F}_p^4 \in \mathbb{R}^{M_4 \times C_{local}}$ have $M_4 = 128$ points and local feature dimension size $C_{local} = 256$. From downsampled points $\boldsymbol{P}^4$ and local features $\boldsymbol{F}_p^4$, we extract global shape descriptor $\boldsymbol{f}_p \in C_{global}$ through a 2-layer MLP followed by max-pooling. We set the global feature dimension size to be $C_{global} = 512$.

## B.2   SEED GENERATOR

Our seed generator produces Patch Seeds with coordinates $\boldsymbol{S} \in \mathbb{R}^{M_{seed} \times 3}$ and features $\boldsymbol{F}_{seed} \in \mathbb{R}^{M_{seed} \times C_{seed}}$. We define our Patch Seeds to have double the resolution of the downsampled partial information they are generated from (i.e., $M_{seed} = 256$) and set the seed feature dimension size to be $C_{seed} = 256$. All of the multi-head attention layers in our seed generator use the same hyperparameters, consisting of 8 heads where each head performs attention on features of dimension size 48. We define our learned embeddings $\boldsymbol{o}_{token} \in \mathbb{R}^C$, $\boldsymbol{e}_{free} \in \mathbb{R}^C$, and $\boldsymbol{e}_{occ} \in \mathbb{R}^C$ to have an embedding size of $C = 384$. We positionally encode downsampled partial points $\boldsymbol{P}^4$, free space points $\boldsymbol{P}_{free}$, and occluded space points $\boldsymbol{P}_{occ}$ via a 2-layer MLP, mapping the points to features of the same dimension size as our learned embeddings. The object's center is predicted by a 3-layer MLP, and the seed offsets which are added to the object center are predicted via a separate 3-layer MLP.

### B.3  COARSE-TO-FINE DECODER

Our decoder uses the upsampling layer proposed by SeedFormer Zhou et al. (2022) for producing a dense completion of an object in a hierarchical fashion. We use a neighborhood size of 20 when computing local attention in the Upsample Transformer of their upsampling layer. For our added global attention layers, we once again use multi-head self-attention with 8 heads. We upsample our completion 3 times, doubling the number of points present in the completion at each upsampling layer, to produce a dense completion of $M_{dense} = 2048$ points.

### B.4  SCENE CONSTRAINTS

Scene constraints are generated from the input partial scan $\boldsymbol{P}_{in} \in \mathbb{R}^{M_{in} \times 3}$ and its estimated surface normals $\boldsymbol{N}_{in} \in \mathbb{R}^{M_{in} \times 3}$. As described in Section C.4, our partial scans are produced by backprojecting depth maps from 10 different viewpoints and resampling the scene to a 2 cm resolution. As we are backprojecting a depth map into a point cloud, we also estimate surface normals, orienting them towards the camera the point cloud was generated from so that the normals are pointing towards "free space". With our partial points and estimated normals, we generate constraint points as $\boldsymbol{P}_{in} \pm \delta \boldsymbol{N}_{in}$, where we set $\delta = 2$ cm.

### B.5  3D INSTANCE SEGMENTATION

For 3D instance segmentation, we retrain Mask3D (Schult et al., 2023) on our ScanWCF dataset, following the training procedure that they use for ScanNet. Training for 600 epochs with a batch size of 3 on scenes with 2 cm voxelization takes approximately 4 days on a single RTX 4090 GPU. We note that the instance segmentations produced by Mask3D are only used during inference. During the training phase of our completion model we use the ground truth partial object instances.

### B.6  MESH RECONSTRUCTION

Meshes of our object completions are only reconstructed at inference time. For this, we directly use NKSR's (Huang et al., 2023) pre-trained *kitchen-sink* model, which has been jointly trained across several object and scene scale datasets.

## C  SCANWCF DATASET

Our proposed ScanWCF dataset is composed of data gathered from the ShapeNet (Chang et al., 2015), ScanNet (Dai et al., 2017), Scan2CAD (Avetisyan et al., 2019a), and ScanARCW (Li et al., 2023a) datasets. All data was obtained directly through publicly available download links on their websites and permission to use the data was granted for datasets which required it.

The ScanWCF dataset consists of 1202 scenes, which are based off scenes from the ScanNet dataset. Our dataset does not contain all 1513 scans from ScanNet as it relies on the background meshes from the ScanARCW dataset, which are only provided for 1274 of the scenes. Additionally, the scenes in our dataset must pass a manual verification process of which 72 out of the 1274 scenes failed. Despite our dataset not containing all the scenes from ScanNet, we reuse the scene id train/test split from the original dataset, resulting in 946 training scenes and 246 test scenes. Objects from 34 different categories from ShapeNet are present across all the scenes in our dataset. While we train each method with all the objects present in the training scenes, we only evaluate the instance scene completion task on the 13 categories which had more than 150 training examples present across all the scenes. We chose this as we observed poor instance segmentation performance across all methods on categories which contained a small number of training examples.

In the following sections, we describe the data generation process for our proposed ScanWCF dataset, which is both Watertight and Collision Free (hence the name ScanWCF).

### C.1  PRE-PROCESSING FOR WATERTIGHTNESS

The objects provided in the ShapeNet dataset are not guaranteed to be watertight, making it difficult to reason about collisions between objects. To address this, we process ShapeNet objects for wa-

tertightness using the method proposed by Wang et al. (2022). This enables us to obtain the signed distance of any point with respect to the object mesh, making collision checking easy. To reason about collisions with scene boundaries (e.g., walls, floor, ceiling), we use the background meshes from the ScanARCW dataset. While the background mesh for each scene aligns with the corresponding scan from ScanNet, some of the background meshes have holes in them where window or door meshes would be placed. Unlike ScanARCW, we leave out the door and window meshes and instead fill the holes in the background meshes to make them watertight.

## C.2 Optimizing scene layout

Our goal is to be able to produce a scene layout containing meshes which closely align with objects in a ScanNet scene while being free of collisions. In order to do so, we begin by initializing our scene as the empty watertight background mesh which is already aligned with a real scan from ScanNet. We then proceed to place one object into the scene at a time, optimizing for both the pose and scale of the object. We make use of the Scan2CAD dataset for deciding which ground truth meshes from ShapeNet to place in the scene, as the dataset has already detected objects in the real ScanNet scenes and matched them to their closest mesh in ShapeNet. Additionally, we initialize each object's pose and scale using the Scan2CAD annotations as they have roughly aligned the synthetic meshes to the real scans. Then we optimize the pose and scale of each object such that the following criteria are best met:

1. **Alignment with partial scan.** We minimize the Chamfer Distance between points sampled on the ground truth mesh and the partial object it was matched to within the real ScanNet scene.

2. **Minimize amount of collisions.** We uniformly sample points on the object mesh and compute their signed distance to the scene mesh (both objects and background). We penalize points which fall inside another object or outside the scene boundaries.

3. **Minimize large changes in scale.** We penalize large deviations in object scale from their initialized value. This prevents the optimization from shrinking the object scale by large amounts to reduce collisions.

4. **Minimize floating objects.** Minimizing the amount of collisions while penalizing changes in scale can lead to the optimization producing objects floating in space. To help prevent this, we sample points on the bottom of the object mesh and compute their signed distance to the scene mesh. We penalize values which deviate from 0, which signifies the object is not resting on a surface.

After an object's pose and scale has been optimized, it is added to the ground truth scene mesh to provide further constraints for the objects which will later be optimized.

## C.3 Manual verification

Not all scene layouts are guaranteed to be free of collisions upon the optimization phase finishing. Therefore, after optimizing for the scene layout, we manually verify that the scene is plausible. If no objects are in collision with each other and the scene looks plausible (e.g., no floating objects), we include the scene in our dataset. If the scene includes some objects which are in collision or are implausible, we manually correct the issue and include the scene in our dataset. If too many collisions still exist after our optimization step, we simply disregard the scene from being included in the dataset.

## C.4 Producing aligned partial scans

After the manual verification phase, we have a set of scenes which serve as the ground truth instance completions for our dataset and now need to generate partial scans which are aligned to them. To generate partial scans of the scene, we render depth maps from different viewpoints and backproject them back into 3D. For a particular scene id, we make use of the camera extrinsics and intrinsics from the corresponding scene id in the ScanNet dataset to render 2D information with. Using PyTorch3D (Ravi et al., 2020), we render depth maps, 2D instance segmentations, and surface normal maps of the scene from each viewpoint. We share some example renderings of a scene in Figure 8a. Upon

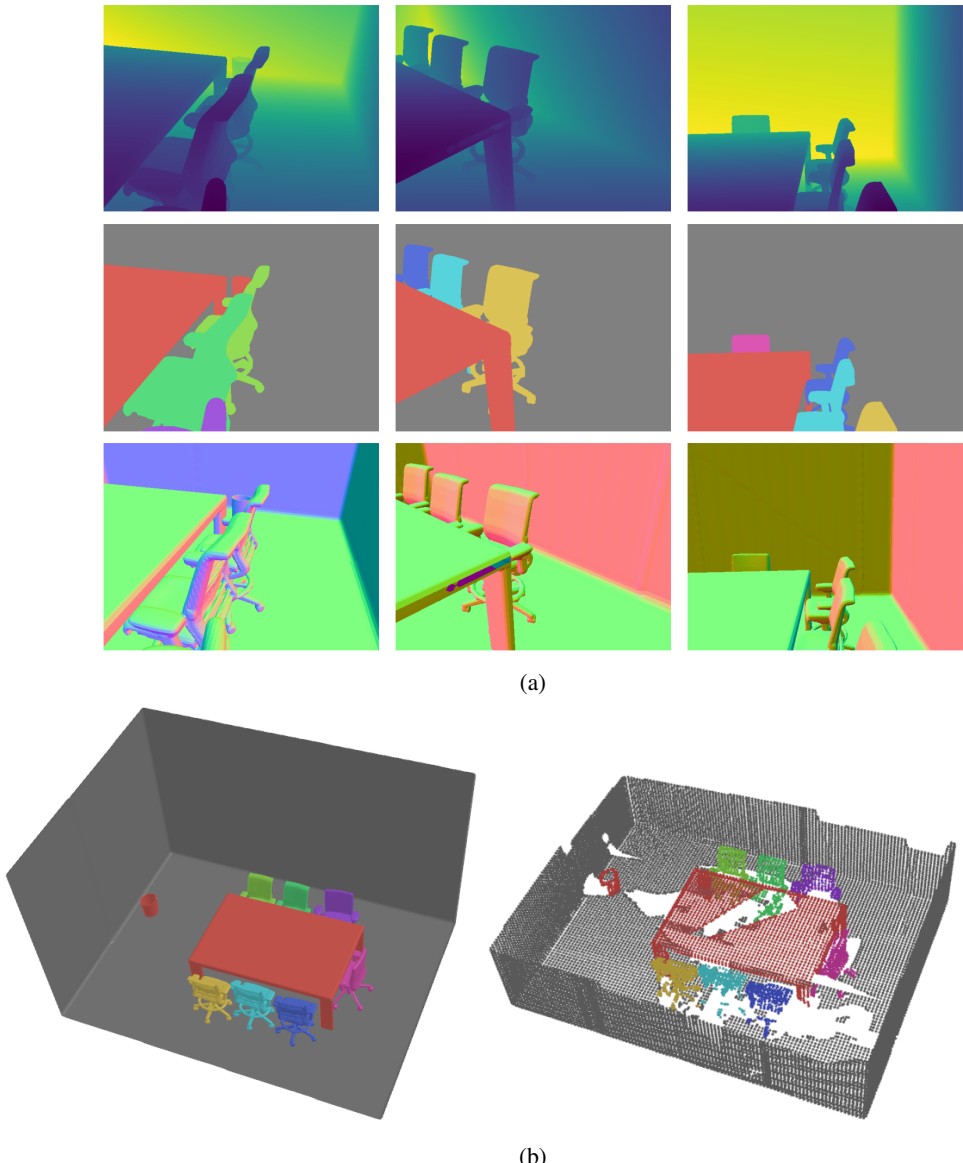

(a)

(b)

Figure 8: (a) We render our ground truth scene meshes into depth maps, instance segmentation maps, and surface normal maps from multiple views. (b) Each scene contains a ground truth labeled scene mesh and an aligned and labeled partial scan constructed from our multi-view renderings of the scene.

rendering 2D information, we select 10 viewpoints per scene and backproject their 2D information into world coordinates, fusing together the different point clouds to produce a partial scan as shown in Figure 8b. While backprojecting the depth map of each viewpoint into a point cloud, we also estimate surface normals and orient them such that they are correctly oriented towards the camera. To keep the number of points in the scene reasonable, we resample each partial scan using grid subsampling with a 2 cm resolution. Finally, To increase the amount of training/test data, for each scene, we choose to generate 2 partial scans using a different subset of 10 ScanNet camera poses.

## D  METRICS

### D.1  SCENE COMPLETION METRICS

For the scene completion task, we have removed the need to predict instance proposals from each method. Instead we provide each method with the ground truth instance segmentations so that each network is only responsible for completing the partial object. In this setting, false positive detections/proposals are not possible and therefore we do not have to rely on the use of mean Average Precision (mAP) as a way to evaluate completion quality. Instead we can evaluate the completion quality of each approach using common metrics from the point cloud completion literature. Each metric used in the scene completion task is defined in the following sections.

#### D.1.1  CHAMFER DISTANCE (CD)

The Chamfer Distance between two point clouds $\boldsymbol{P} \in \mathbb{R}^{N \times 3}$ and $\boldsymbol{Q} \in \mathbb{R}^{M \times 3}$ can be defined as:

$$d^{CD}(\boldsymbol{P}, \boldsymbol{Q}) = \frac{1}{|\boldsymbol{P}|} \sum_{\boldsymbol{x} \in \boldsymbol{P}} \min_{\boldsymbol{y} \in \boldsymbol{Q}} \|\boldsymbol{x} - \boldsymbol{y}\|_2^2 + \frac{1}{|\boldsymbol{Q}|} \sum_{\boldsymbol{y} \in \boldsymbol{Q}} \min_{\boldsymbol{x} \in \boldsymbol{P}} \|\boldsymbol{x} - \boldsymbol{y}\|_2^2 \tag{5}$$

We use Chamfer Distance as a measure of completion quality. For evaluation, we sample 2048 points per predicted object and concatenate each object's point set into a single scene point cloud. We do the same procedure for the ground truth objects, uniformly sampling the 2048 points from each object mesh. Chamfer Distance is then measured between the predicted scene point cloud and ground truth scene point cloud using Equation 5. Numbers reported in tables have been scaled by a factor of $10^3$.

#### D.1.2  ONE-SIDED CHAMFER DISTANCE

Equation 5 measures the bi-directional (or symmetric) Chamfer Distance. To evaluate how well the completion respects the partial input, we instead measure the One-sided Chamfer Distance. The One-sided Chamfer Distance between two point clouds $\boldsymbol{P} \in \mathbb{R}^{N \times 3}$ and $\boldsymbol{Q} \in \mathbb{R}^{M \times 3}$ can be defined as:

$$d^{OCD}(\boldsymbol{P}, \boldsymbol{Q}) = \frac{1}{|\boldsymbol{P}|} \sum_{\boldsymbol{x} \in \boldsymbol{P}} \min_{\boldsymbol{y} \in \boldsymbol{Q}} \|\boldsymbol{x} - \boldsymbol{y}\|_2^2 \tag{6}$$

We measure the One-sided Chamfer Distance between the 2 cm resolution partial scan excluding the background (i.e., points on the walls, floor, ceiling) and our predicted scene point cloud. Numbers reported in tables have been scaled by a factor of $10^3$.

#### D.1.3  UNIDIRECTIONAL HAUSDORFF DISTANCE (UHD)

One-sided Chamfer Distance measures the average deviation of the partial reconstruction from the partial input. We additionally measure the maximum deviation of the partial reconstruction from the partial input using the Unidirectional Hausdorff Distance (UHD). The Unidirectional Hausdorff Distance between point clouds $\boldsymbol{P} \in \mathbb{R}^{N \times 3}$ and $\boldsymbol{Q} \in \mathbb{R}^{M \times 3}$ can be defined as:

$$d^{UHD}(\boldsymbol{P}, \boldsymbol{Q}) = \max_{\boldsymbol{x} \in \boldsymbol{P}} \min_{\boldsymbol{y} \in \boldsymbol{Q}} \|\boldsymbol{x} - \boldsymbol{y}\|_2 \tag{7}$$

Similar to One-Sided Chamfer Distance, we measure UHD between the 2 cm resolution partial scan with background removed and our predicted scene point cloud. Numbers reported in tables have been scaled by a factor of $10^3$.

#### D.1.4  COLLISION METRIC (COL)

We measure completion plausibility by how badly predicted completions collide with each other or the scene boundaries. Let the watertight background mesh (i.e., walls, floor, ceiling) be denoted as $\mathcal{B}$ and the set of predicted watertight object meshes be denoted as $\{\mathcal{M}_1, ..., \mathcal{M}_n\}$. We define $SDF(\mathcal{M}, p)$ to represent the signed distance of an arbitrary 3D point $p$ to its closest point on the surface of a mesh $\mathcal{M}$, where $SDF(\mathcal{M}, p) < 0$ implies the point $p$ falls inside mesh $\mathcal{M}$. Additionally, we denote our set of point clouds which we reconstructed meshes from as $\{\boldsymbol{C}_1, ..., \boldsymbol{C}_n\}$

(for RfD-Net and DIMR we sample $2048$ points uniformly from a mesh $\mathcal{M}_i$ to produce point cloud completion $C_i$). Each object point cloud $C_i$ is composed of a set of 3D points, where we denote a point being in the point cloud as $p \in C_i$. Now we can define the collision metric for a scene as:

$$COL = -\frac{1}{n} \sum_{i=1}^{n} \frac{1}{|C_i|} \sum_{p \in C_i} \left( \min(0, \ -SDF(\mathcal{B}, p)) + \sum_{\substack{j=1 \\ i \neq j}}^{n} \min(0, SDF(\mathcal{M}_j, p)) \right)$$

If no points in the scene completion are violating scene constraints (i.e., penetrating into other objects or the scene boundaries), the collision metric is 0. Otherwise, the collision metric is equal to the average of how far each point has penetrated into an object or extended outside the scene boundaries.

## D.2 INSTANCE SCENE COMPLETION METRICS

To evaluate our proposed approach on the joint task of instance segmentation and object completion, we use the metrics and setup proposed by Tang et al. (2022). In particular, we compute the 3D detection mean Average Precision (mAP) for each metric defined in the following sections. For computing average precision, the PASCAL VOC 2007 11-point interpolation method is used.

### D.2.1 INTERSECTION OVER UNION (IoU)

Intersection over Union is a voxel-based approach for evaluating completion quality. It measures the predicted completion's voxel occupancy against the ground truth completion's voxel occupancy. To compute it, both predicted meshes and ground truth meshes are first voxelized with a fixed voxel size of $4.7$ cm. Then Intersection over Union is computed as the $\frac{\text{Volume of overlap}}{\text{Volume of union}}$ with regards to voxel occupancy.

### D.2.2 CHAMFER DISTANCE (CD)

Chamfer Distance provides a point-based evaluation of completion quality. It measures the distance from points sampled on the surface of the predicted mesh to points sampled on the ground truth mesh. To generate point sets, we uniformly sample $4096$ points from both the predicted mesh and the ground truth mesh. Chamfer Distance is then computed using Equation 5.

### D.2.3 LIGHT FIELD DISTANCE (LFD)

Light Field Distance is a visual similarity metric for meshes. The main idea is that if the predicted complete mesh is similar to the ground truth mesh then it should look similar from all viewpoints. To compute LFD, each predicted mesh and ground truth mesh is rendered into 2D images from multiple viewpoints and encoded into a light field descriptor that is used for measuring distances between meshes. We refer readers to the work by Chen et al. (2003) for a description of the light field descriptor.

### D.2.4 POINT COVERAGE RATIO (PCR)

Point Coverage Ratio measures how well the completed mesh aligns with the partial input. The Point Coverage Ratio between a point set $P \in \mathbb{R}^{N \times 3}$ and a mesh $\mathcal{M}$ can be defined as:

$$PCR(P, \mathcal{M}) = \frac{1}{|P|} \sum_{x \in P} \mathbb{1}_{\{\text{dist}(x, \mathcal{M}) < \tau_{pcr}\}} \tag{8}$$

where $\mathbb{1}$ is the indicator function, $\text{dist}(x, \mathcal{M})$ is the distance from the point $p$ to the surface of mesh $\mathcal{M}$, and $\tau_{pcr}$ is a distance threshold (we set $\tau_{pcr} = 0.047$ for evaluation).

## D.3 SURFACE NORMAL METRICS

In section E.8, we evaluate the quality of our surface normal predictions needed to reconstruct a mesh. Rather than directly evaluate accuracy of the predicted surface normals, we evaluate the

quality of the reconstructed meshes produced using NKSR (Huang et al., 2023) with our point cloud and surface normals. To evaluate object mesh quality, we use the metrics and setup proposed by Park et al. (2019).

### D.3.1 CHAMFER DISTANCE (CD)

Chamfer Distance is used to evaluate overall shape quality. In particular, we sample $30,000$ points on both the predicted and ground truth meshes and compute the Chamfer Distance between the two point sets using Equation 5. Here a lower value indicates a better match to the ground truth shape.

### D.3.2 MESH COMPLETION

Mesh completion evaluates how well the ground truth surface is covered by the predicted mesh reconstruction. In particular, we compute the percent of points sampled from the ground truth surface whose distance to the predicted mesh reconstruction is within a threshold $\tau_{comp}$ (we set $\tau_{comp} = 0.01$ following Park et al. (2019)). We compute the mesh completion metric over 1000 points uniformly sampled from the ground truth mesh. A higher score indicates better coverage of the ground truth surface by the predicted mesh reconstruction.

### D.3.3 MESH ACCURACY

Mesh accuracy evaluates how close points on the predicted mesh surface are to the surface of the ground truth mesh. In particular, we find the minimum distance threshold $\tau_{acc}$ such that $90\%$ of points sampled from the surface of the predicted mesh are within distance $\tau_{acc}$ to the ground truth surface. To compute mesh accuracy, we sample 1000 points from the predicted mesh reconstruction and compute each points distance to the ground truth surface. A lower score indicates better accuracy to the ground mesh.

## E RESULTS

In this section we share more results of our scene completion method.

### E.1 RUNTIME

On an RTX 4090 GPU, our completion model takes an average of $0.104$ seconds to complete a scene containing an average of 9 objects in it. In other words, our method can complete about 86 objects per second.

### E.2 INSTANCE SCENE COMPLETION

In Table 5 and Table 6, we share per class Average Precision (AP) scores for the Intersection over Union (IoU) metric at thresholds $0.25$ and $0.5$, respectively. Note that at both thresholds our method outperforms RfD-Net and DIMR across almost all categories. In particular, we observe a significant gap in the performance of our method compared to previous approaches when evaluating at the more difficult threshold (Table 6).

In Table 7 and Table 8, we show per class AP scores for the Chamfer Distance (CD) metric at threshold $0.1$ and $0.047$, respectively. Both RfD-Net and DIMR suffer large drops in performances when evaluating on the more difficult threshold (Table 8). On the other hand, the drops in performance, if any, for our method are much smaller.

Along with CD, we evaluate completion quality using the Light Field Distance (LFD) metric which measures visual similarity between meshes. In Table 9 and Table 10, we see that RfD-Net and DIMR behave similarly to how they do with CD and IoU, achieving decent performance on easier thresholds but very low scores at the more difficult thresholds. This trend occurring across all three metrics suggests that previous approaches can only produce completions which are somewhat similar in shape to the ground truth objects. On the other hand, we see our method obtains a much higher LFD score on the more challenging threshold, which is similar to how we perform on CD and IoU

compared to previous works, suggesting that our method produces completions that more accurately represent the geometry of the objects in the scene.

To evaluate partial reconstruction quality, in Table 11 and Table 12, we share per class AP scores on the Point Coverage Ratio (PCR) metric. Our method observes almost no drop in performance between the easier and more difficult threshold across all categories, demonstrating that our completions align with the partial scans with high fidelity.

Finally, in Figure 9 and Figure 10 we share more qualitative comparisons against previous works RfD-Net and DIMR on the instance scene completion task.

### E.3   Scene completion

In Figure 11, we share qualitative results when the ground truth instances have been supplied to each method. Despite having the ground truth instance information, RfD-Net still produces low-quality completions of chairs and tables while DIMR fails to complete or even reconstruct the observed inputs in many cases. Additionally, we see RfD-Net tends to produce many collisions between completions. On the other hand, our approach respects the partial input well and completes the fine-grained geometry that is missing from the scans while avoiding collisions between predictions.

### E.4   Analyzing the effects of object scan incompleteness

In Figure 12, we further investigate how the completion quality of our method varies under different levels of incompleteness in the partial input. In Figure 12a, we present a histogram breaking down the object instances in the test set of our dataset by how much of the complete shape is present in the partial object scan. We find that a majority of our partial object scans contain between $30 - 60\%$ of the complete geometry before being input to our completion model. In Figures 12b - 12d, we plot the average completion metrics for various methods for each bin in our histogram. We find that our method outperforms the baseline approaches at each completeness level by a large margin, which is consistent with the large gap in performance observed in our main results shown in Table 1 and Table 2. However, we do observe that RfD-Net and DIMR do outperform our method for extremely sparse inputs (e.g., when only $0 - 10\%$ of the object is present in the partial scan).

### E.5   Analyzing the effects of imperfect instance segmentation

In Figure 13, we visualize some example completions when there are errors present in the instance segmentation predictions produced by Mask3D. In the top four rows, we show examples where Mask3D produced segmentations that missed large or important parts of the partial object instance. The regions missed by Mask3D contain important cues for the true geometry and size of the object, and the lack of this information leads our model to produce a completion which is different from the ground truth completion. In the bottom two rows, we show examples where Mask3D incorrectly segments two objects that are side by side as a single object. In this scenario, our completion model has no knowledge that there are actually two objects present and instead completes both objects together as if it were one.

### E.6   Generalization to real scans

In Figure 14, we share some example completions which demonstrate our methods ability to generalize to partial objects from real scans in the ScanNet dataset. While we sample object instances from categories we have trained on (e.g., chair, table, trash bin), all the object instances shown are completely novel as the ScanNet object instances have no overlap with objects in the ShapeNet dataset. Moreover, the partial scans from ScanNet are considerably less clean than the partial scans from our dataset, yet our method is still able to produce plausible completions of the missing regions of the objects.

### E.7   Ablations

In Figure 15 we present a qualitative comparison of our scene completion model with and without pre-training. While our method can outperform previous approaches without pre-training the object

completion model, we find that pre-training is important for reasoning about the missing geometry when entire parts of an object are missing from the partial scan. In the first scene of Figure 15, we see that without pre-training our completion model is able to represent the thin legs of the chair when some of the legs are observed in the partial scan (e.g., cyan and green chairs), but produces a solid circular base instead of thin legs when the entire base of the chair is missing (e.g., blue, purple, and pink chairs). On the other hand, when we pre-train our object completion model we are able to recover the individual legs of the chair despite not observing any part of it due to the model having a prior over objects which it can fall back on when needing to hallucinate the missing structures of largely occluded objects. Furthermore, we find that pre-training helps clean up noisy predictions such as the dark blue chair in the third scene of Figure 15.

### E.8 SURFACE NORMAL ESTIMATION

In Figure 16, we share a qualitative comparison of our completions and estimated surface normals compared to the ground truth meshes and surface normals. We find that our method does a good job at recovering the complete geometry of the scene even for small details such as the handles on drawers of the bathroom sink in the first scene or the thin cross bars on the legs of the chairs in the last scene.

For mesh reconstruction, NKSR (Huang et al., 2023) typically uses a PCA-based plane fitting technique for estimating normals of a point cloud. In Table 13 and Figure 17, we show that normals estimated using PCA-based plane fitting suffer from poor reconstruction quality on our completions. We find that varying the neighborhood size used in plane fitting does not seem to help improve the reconstructions. Instead the normals produced by our normal estimation module lead to significantly better reconstructions, outperforming PCA-based normal estimation across all metrics. In Figure 17, we show that PCA-based plane fitting suffers from inaccurate normal estimation in the presence of noise in the completion or on thin structures such as the legs of chairs, leading to artifacts in the reconstructions. Alternatively, our estimated normals are oriented correctly despite noise being present or the completion containing thin structures, producing much better reconstructions with NKSR.

### E.9 FAILURE CASES

In Figure 18, we share some example failure cases of our scene completion model. We find that our method occasionally suffers from collisions between predictions when two objects are up against each other but there are no scene constraint points suggesting a separation between the two objects. An example of this can be seen in the left most scene of Figure 18, where the partial scan (black points) contains no information on how far the blue drawer should extend to the right, causing it to overextend into the yellow drawer. While our object completion model does include information about the scene when producing a completion, it only considers observed information present in the partial scan. This means our model cannot reason about avoiding collisions in regions of space that will be filled by another object's completion.

Table 5: **IoU@0.25** Per class AP scores for mesh quality based on occupancy against ground truth voxels using Intersection over Union (IoU) with threshold of 0.25. † denotes our scene completion model results without pre-training.

| | Table | Chair | Bookshelf | Sofa | Lamp | Trash Bin | File Cabinet | Bag | Cabinet | Bed | Display | Bathtub | Printer |
|---|---|---|---|---|---|---|---|---|---|---|---|---|---|
| RfD-Net | 22.79 | 45.75 | 41.90 | 6.97 | 30.37 | 70.83 | **13.95** | **69.81** | 31.21 | 12.34 | 76.80 | **62.66** | 18.79 |
| DIMR | 56.06 | 80.16 | 55.04 | 26.59 | 0.00 | 63.83 | 0.00 | 0.00 | 26.45 | 44.60 | 69.25 | 17.79 | 4.55 |
| Ours † | 70.74 | 86.75 | **66.08** | 61.24 | **72.33** | **77.89** | 11.06 | 63.40 | 49.56 | 61.97 | 78.76 | 62.25 | **43.80** |
| Ours | **71.71** | **86.86** | 65.84 | **68.97** | 72.33 | 77.89 | 11.06 | 63.40 | **49.79** | **63.30** | **79.31** | 62.25 | 43.39 |

Table 6: **IoU@0.5** Per class AP scores for mesh quality based on occupancy against ground truth voxels using Intersection over Union (IoU) with threshold of 0.5. † denotes our scene completion model results without pre-training.

|  | Table | Chair | Bookshelf | Sofa | Lamp | Trash Bin | File Cabinet | Bag | Cabinet | Bed | Display | Bathtub | Printer |
|---|---|---|---|---|---|---|---|---|---|---|---|---|---|
| RfD-Net | 1.81 | 0.46 | 2.19 | 0.09 | 2.27 | 32.08 | 0.65 | 29.55 | 3.03 | 0.20 | 11.76 | 9.88 | 0.30 |
| DIMR | 11.13 | 6.57 | 11.02 | 1.81 | 0.00 | 34.45 | 0.00 | 0.00 | 0.57 | 0.81 | 31.72 | 14.56 | 0.00 |
| Ours † | 37.56 | 72.16 | 45.95 | 29.59 | **59.33** | 75.63 | **10.39** | 53.98 | 31.69 | 10.88 | 64.43 | 51.78 | **43.39** |
| Ours | **39.63** | **74.22** | **47.61** | **29.86** | 54.34 | **76.03** | 10.06 | 53.98 | **37.22** | **31.14** | **66.64** | **61.01** | 36.36 |

Table 7: **CD@0.1** Per class AP scores for mesh quality based on distances between mesh surfaces using Chamfer Distance (CD) with threshold of 0.1. † denotes our scene completion model results without pre-training.

|  | Table | Chair | Bookshelf | Sofa | Lamp | Trash Bin | File Cabinet | Bag | Cabinet | Bed | Display | Bathtub | Printer |
|---|---|---|---|---|---|---|---|---|---|---|---|---|---|
| RfD-Net | 58.39 | 88.28 | **62.66** | 49.33 | 44.71 | 74.80 | **23.30** | 69.84 | 57.30 | 63.28 | 85.74 | **74.44** | **49.78** |
| DIMR | 70.49 | **89.32** | 58.53 | 50.37 | 0.00 | 64.05 | 0.00 | 0.00 | 36.30 | 59.82 | 80.31 | 18.18 | 7.57 |
| Ours † | 71.25 | 86.89 | 57.02 | **61.46** | **72.33** | 77.89 | 11.06 | 63.40 | 49.59 | 63.31 | 79.31 | 61.93 | 43.80 |
| Ours | **71.49** | 86.92 | 57.60 | **61.46** | **72.33** | 77.89 | 11.06 | 63.40 | 49.36 | 62.91 | 79.95 | 62.25 | 43.80 |

Table 8: **CD@0.047** Per class AP scores for mesh quality based on distances between mesh surfaces using Chamfer Distance (CD) with threshold of 0.047. † denotes our scene completion model results without pre-training.

|  | Table | Chair | Bookshelf | Sofa | Lamp | Trash Bin | File Cabinet | Bag | Cabinet | Bed | Display | Bathtub | Printer |
|---|---|---|---|---|---|---|---|---|---|---|---|---|---|
| RfD-Net | 14.55 | 36.07 | 23.12 | 3.13 | 25.63 | 68.78 | **11.25** | 69.12 | 22.54 | 2.27 | 67.51 | 56.85 | 6.32 |
| DIMR | 21.19 | 66.61 | 34.11 | 9.35 | 0.00 | 63.36 | 0.00 | 0.00 | 17.66 | 1.52 | 67.09 | 17.39 | 4.55 |
| Ours † | 57.46 | 85.64 | 46.95 | **45.94** | 70.27 | 77.08 | 10.39 | 63.40 | 40.54 | 38.89 | 68.36 | **61.93** | **43.80** |
| Ours | **59.26** | **86.00** | **48.55** | 45.71 | **70.27** | 77.00 | 10.39 | 63.40 | **41.74** | **51.06** | **77.20** | **61.93** | 36.36 |

Table 9: **LFD@5000** Per class AP scores for mesh quality based on visual appearance using Light Field Distance (LFD) with threshold of 5000. † denotes our scene completion model results without pre-training.

|  | Table | Chair | Bookshelf | Sofa | Lamp | Trash Bin | File Cabinet | Bag | Cabinet | Bed | Display | Bathtub | Printer |
|---|---|---|---|---|---|---|---|---|---|---|---|---|---|
| RfD-Net | 12.55 | 4.73 | 53.75 | 40.56 | 0.00 | 60.80 | **27.65** | 68.67 | 53.97 | 52.93 | 22.45 | **68.63** | 42.61 |
| DIMR | 24.76 | 10.30 | **61.37** | 53.99 | 0.00 | 50.59 | 0.00 | 0.00 | 44.69 | 40.19 | 23.20 | 26.65 | 17.53 |
| Ours † | 50.12 | 59.77 | 46.54 | **60.37** | **36.57** | 73.07 | 18.19 | 63.40 | 50.55 | 59.15 | 54.43 | 60.81 | **43.39** |
| Ours | **51.15** | **64.79** | 55.41 | 58.67 | 27.75 | 60.30 | 18.71 | 63.16 | 52.14 | **62.80** | 54.41 | 61.15 | 36.36 |

Table 10: **LFD@2500** Per class AP scores for mesh quality based on visual appearance using Light Field Distance (LFD) with threshold of 2500. † denotes our scene completion model results without pre-training.

|  | Table | Chair | Bookshelf | Sofa | Lamp | Trash Bin | File Cabinet | Bag | Cabinet | Bed | Display | Bathtub | Printer |
|---|---|---|---|---|---|---|---|---|---|---|---|---|---|
| RfD-Net | 1.81 | 0.43 | 30.54 | 4.85 | 0.00 | 29.24 | **22.94** | 26.93 | 33.67 | 12.73 | 0.12 | 24.09 | 3.09 |
| DIMR | 0.87 | 3.03 | **31.68** | 13.95 | 0.00 | 5.82 | 0.00 | 0.00 | 32.85 | 7.38 | 0.16 | 17.22 | 1.52 |
| Ours † | 24.13 | 28.40 | 15.96 | 42.35 | **5.05** | 19.50 | 13.63 | 13.09 | 39.01 | 22.28 | 26.49 | 33.08 | **36.36** |
| Ours | **28.72** | **32.47** | 23.35 | **45.74** | 3.64 | 28.82 | 12.25 | **27.30** | 39.21 | 28.76 | 32.17 | 38.55 | 36.36 |

Table 11: **PCR@0.5** Per class AP scores for point-to-mesh mapping quality using point coverage ratio (PCR). † denotes our scene completion model results without pre-training.

|  | Table | Chair | Bookshelf | Sofa | Lamp | Trash Bin | File Cabinet | Bag | Cabinet | Bed | Display | Bathtub | Printer |
|---|---|---|---|---|---|---|---|---|---|---|---|---|---|
| RfD-Net | 47.76 | **88.79** | 60.50 | 34.58 | 43.87 | 72.11 | **22.69** | 69.57 | 58.00 | 67.84 | 79.43 | **74.79** | **54.38** |
| DIMR | **76.87** | 80.67 | 64.08 | 59.07 | 0.00 | 63.95 | 0.00 | 0.00 | 37.22 | 70.70 | 77.33 | 26.65 | 7.27 |
| Ours † | 73.31 | 86.97 | **66.42** | **68.98** | **72.33** | 77.89 | 11.06 | 63.40 | 50.68 | 70.94 | 80.37 | 62.17 | 43.80 |
| Ours | 73.26 | 86.97 | 66.38 | 68.97 | **72.33** | 77.89 | 11.06 | 63.40 | 50.68 | **70.94** | 80.37 | 62.17 | 43.80 |

Table 12: **PCR@0.75** Per class AP scores for point-to-mesh mapping quality using point coverage ratio (PCR). † denotes our scene completion model results without pre-training.

|  | Table | Chair | Bookshelf | Sofa | Lamp | Trash Bin | File Cabinet | Bag | Cabinet | Bed | Display | Bathtub | Printer |
|---|---|---|---|---|---|---|---|---|---|---|---|---|---|
| RfD-Net | 34.72 | 59.53 | 30.56 | 15.64 | 25.32 | 58.03 | **13.64** | 67.85 | 47.60 | 28.88 | 78.33 | **64.10** | 26.47 |
| DIMR | 54.38 | 65.68 | 26.90 | 35.96 | 0.00 | 61.95 | 0.00 | 0.00 | 31.00 | 38.65 | 68.02 | 17.80 | 0.04 |
| Ours † | **70.80** | **86.33** | 65.25 | **68.63** | **72.33** | 77.89 | 11.06 | 63.16 | **49.97** | 63.23 | 80.37 | 62.17 | **43.80** |
| Ours | 70.32 | 86.13 | **65.55** | **68.63** | **72.33** | 77.89 | 11.06 | 63.15 | 49.90 | **63.44** | 80.37 | 62.17 | **43.80** |

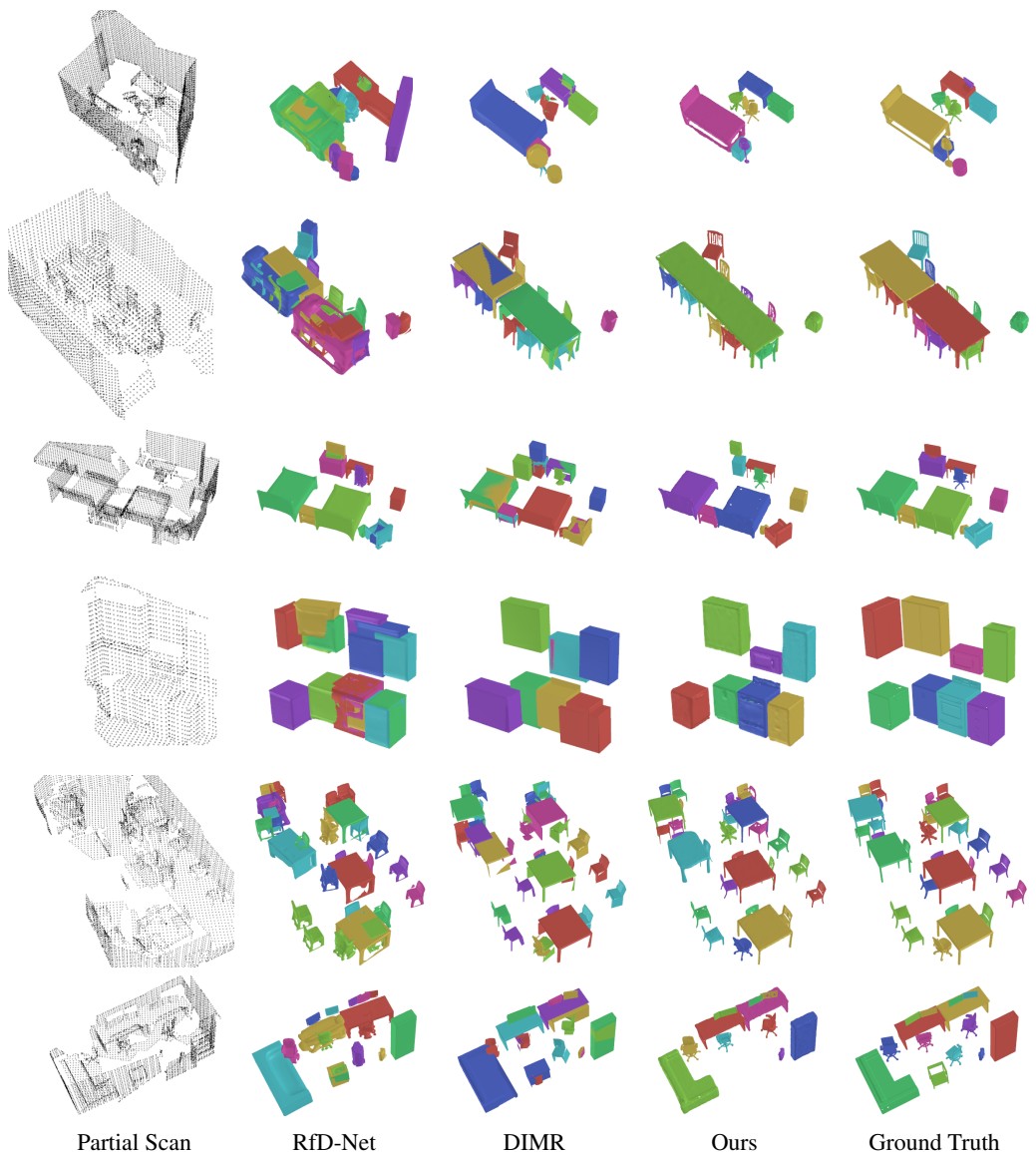

Partial Scan      RfD-Net      DIMR      Ours      Ground Truth

Figure 9: Qualitative comparison on the instance scene completion task.

Table 13: Evaluation of mesh reconstruction quality using our predicted surface normals vs. a PCA-based normal estimation.

|  | CD ($\downarrow$) | Mesh Comp. ($\uparrow$) | Mesh Acc. ($\downarrow$) |
| --- | --- | --- | --- |
| PCA (k=16) | 28.83 | 0.327 | 0.062 |
| PCA (k=32) | 28.58 | 0.331 | 0.063 |
| PCA (k=48) | 28.51 | 0.321 | 0.062 |
| Ours | **21.12** | **0.508** | **0.046** |

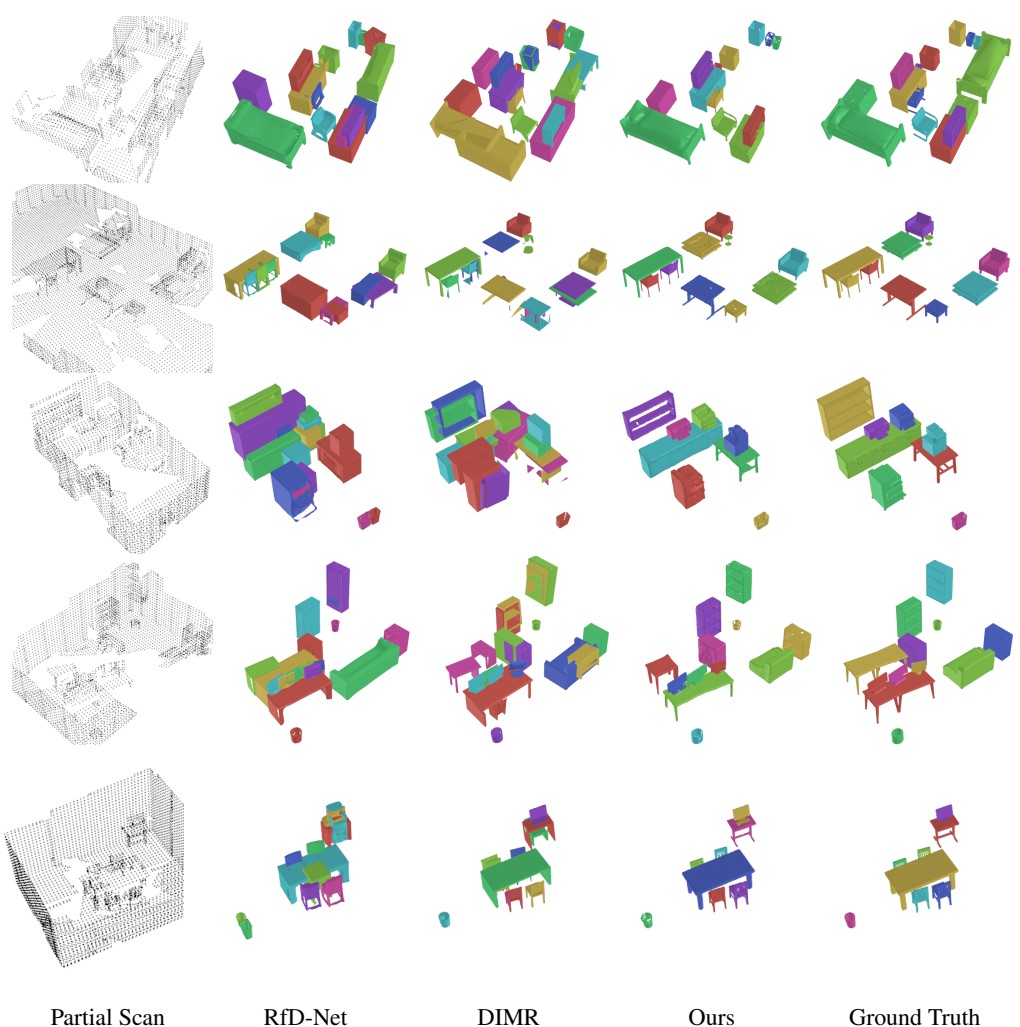

Partial Scan      RfD-Net      DIMR      Ours      Ground Truth

Figure 10: More qualitative comparisons on the instance scene completion task.

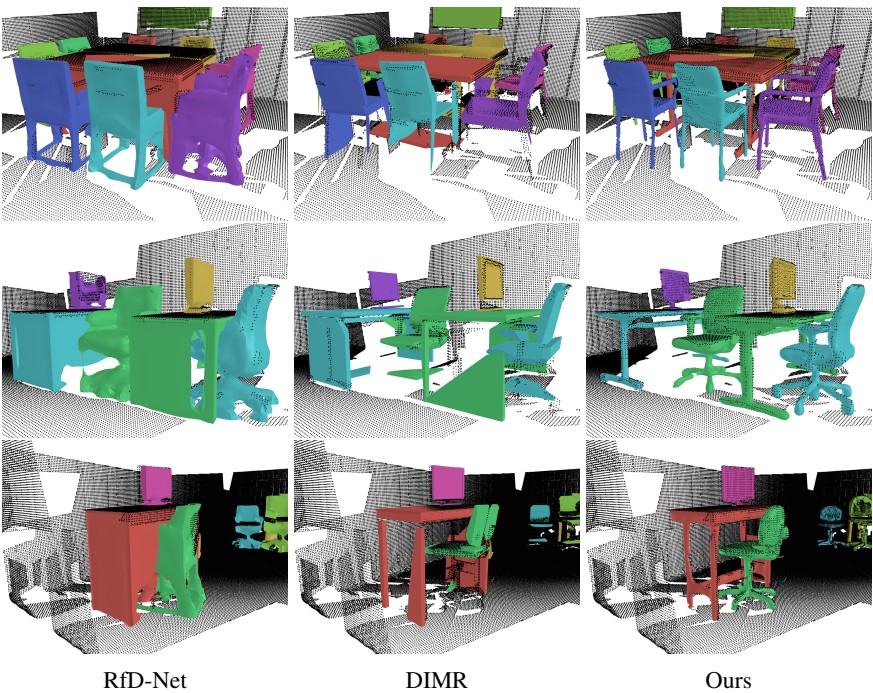

RfD-Net      DIMR      Ours

Figure 11: Qualitative results on the scene completion task when ground truth instance information is provided.

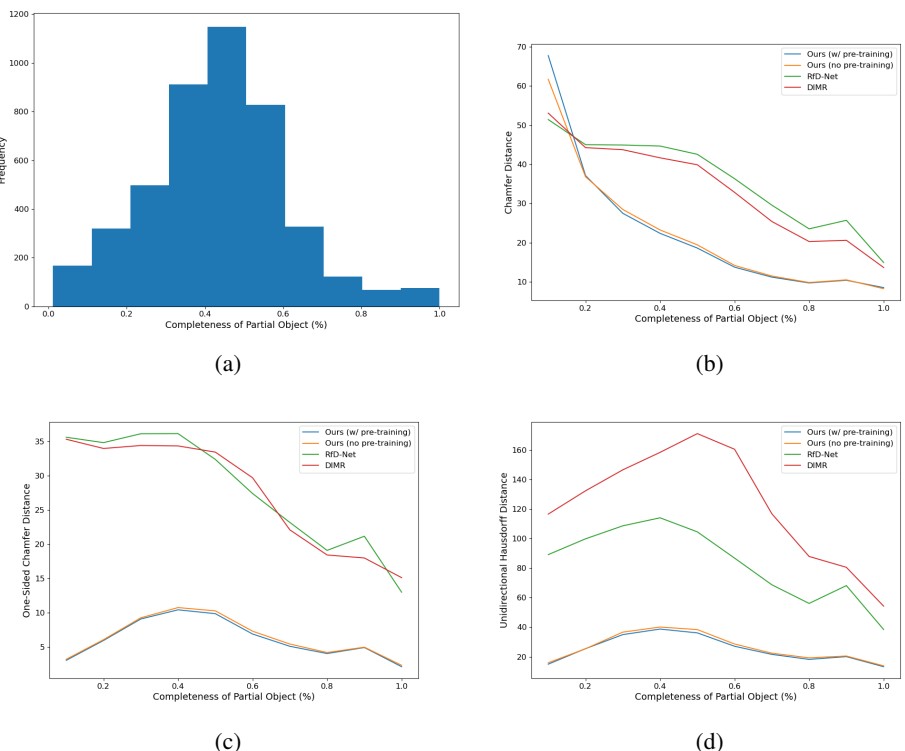

(a)                                                              (b)

(c)                                                              (d)

Figure 12: (a) Histogram breaking down the objects in our test dataset based on how much of the complete shape is present in the partial object scan. (b) Comparison of average Chamfer Distance for various methods for each bin in our histogram. (c) Comparison of average One-Sided Chamfer Distance for various methods for each bin in our histogram. (d) Comparison of average Unidirectional Hausdorff Distance for various methods for each bin in our histogram. We note that each data point is an average over all the partial objects that fall into a bin of our histogram (e.g., the data points at x=1.0 are really the average metric on completions over all partial objects which fell into histogram bin [0.9, 1.0]).

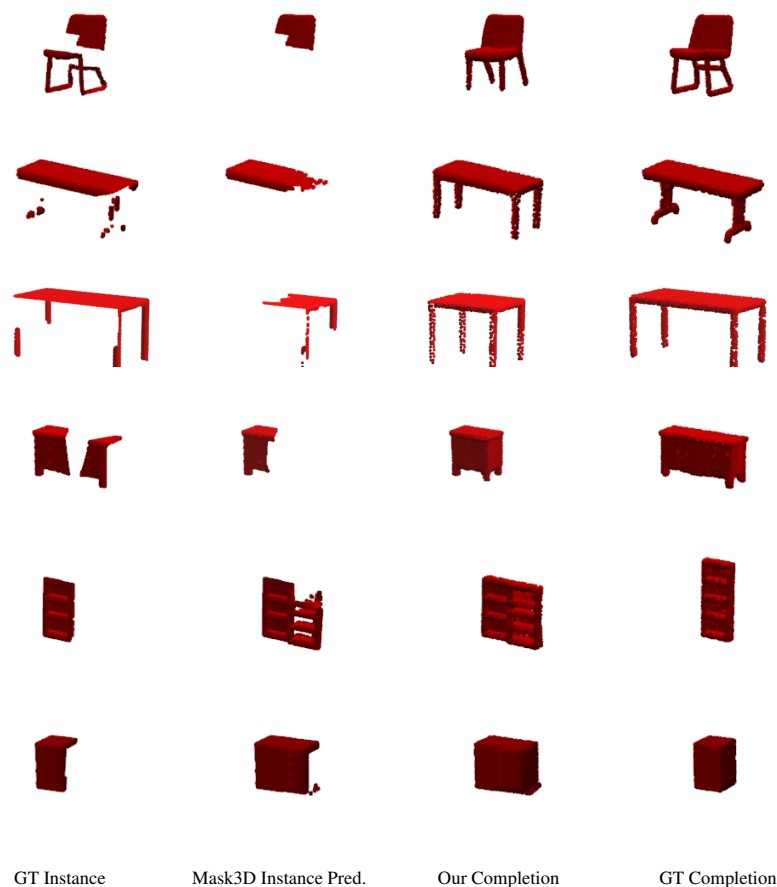

GT Instance          Mask3D Instance Pred.          Our Completion          GT Completion

Figure 13: Example completions on imperfect instance segmentation predictions. Errors in the instance segmentations produced by Mask3D can lead to our method producing incorrect completions.

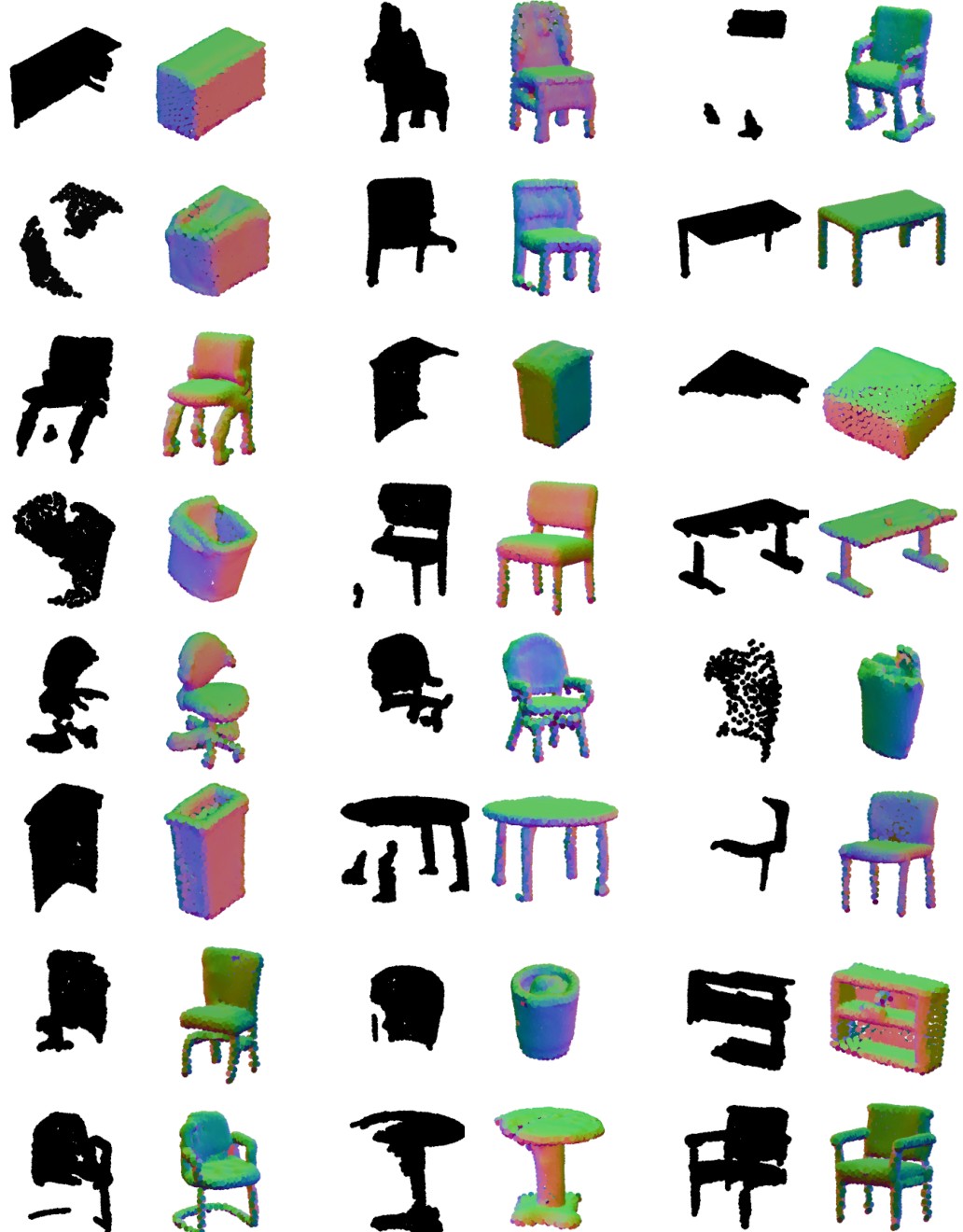

Figure 14: Example completions when generalizing to partial objects from real scans in the ScanNet dataset (Dai et al., 2017).

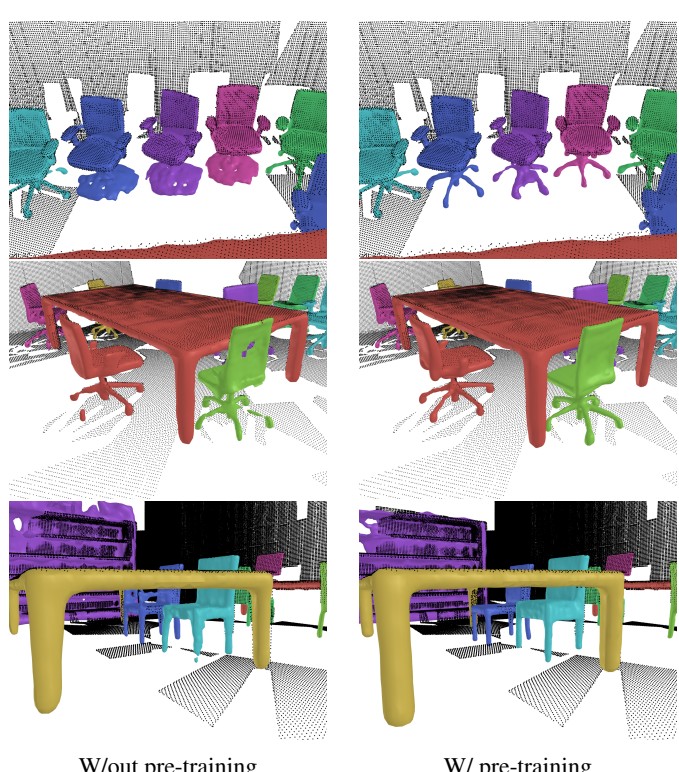

W/out pre-training          W/ pre-training

Figure 15: Qualitative comparison of our scene completion model with and without pre-training.

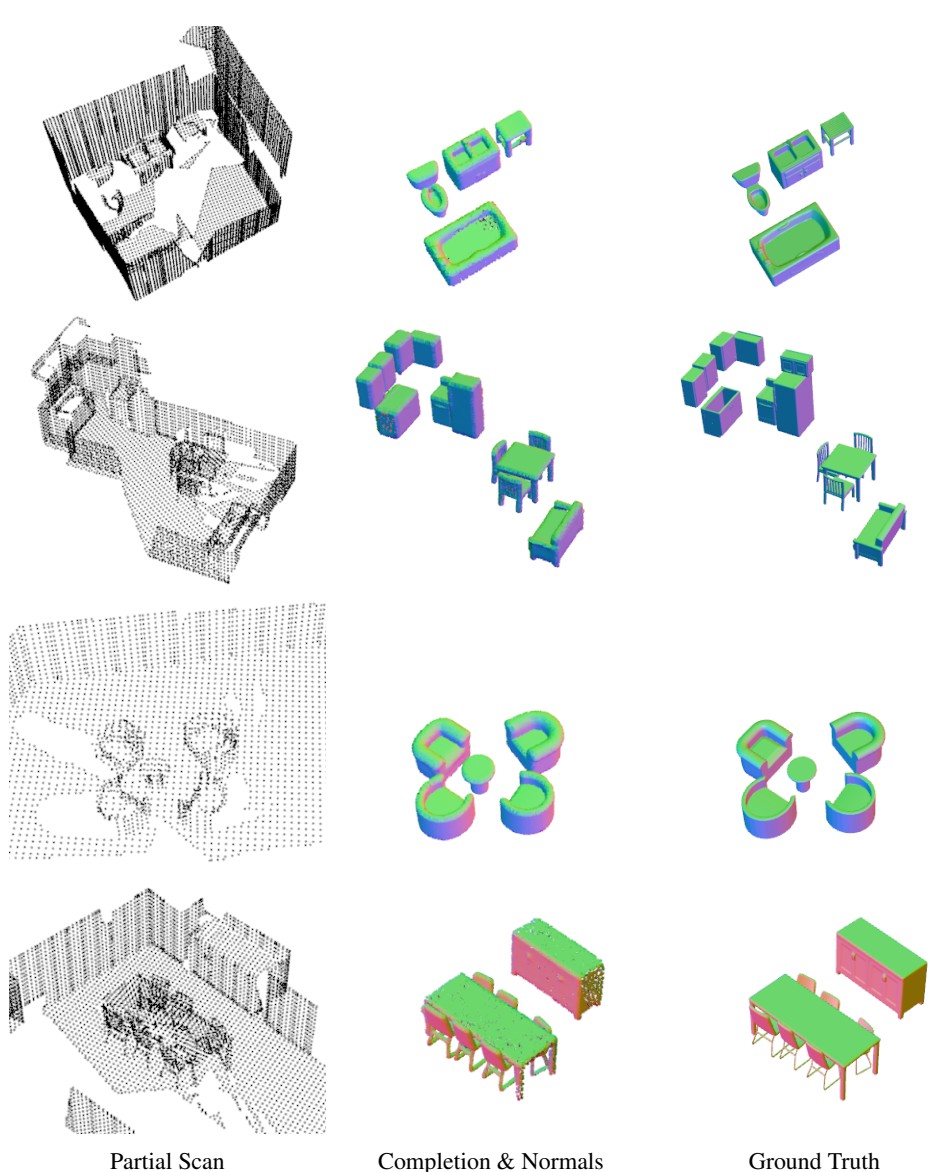

Partial Scan        Completion & Normals        Ground Truth

Figure 16: Qualitative results of our predicted completions and surface normals.

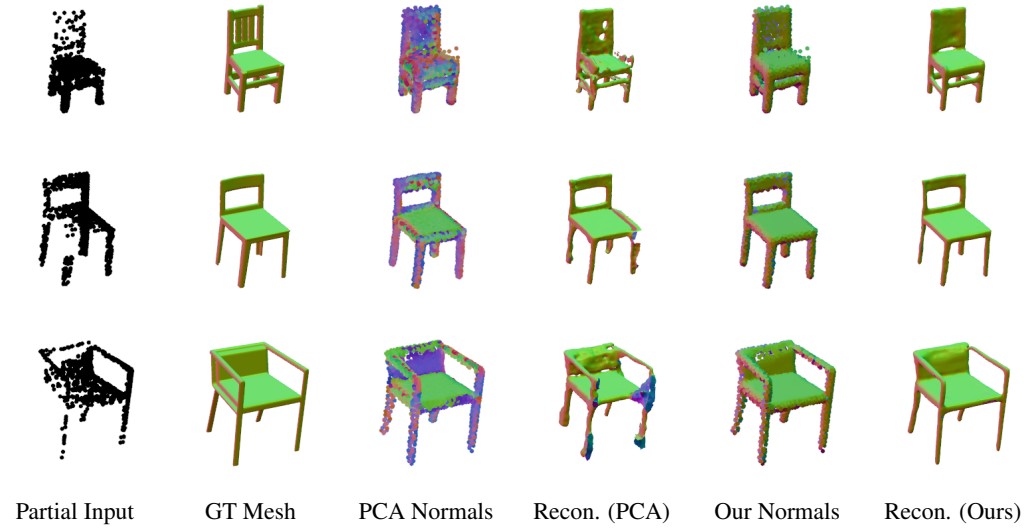

| Partial Input | GT Mesh | PCA Normals | Recon. (PCA) | Our Normals | Recon. (Ours) |

Figure 17: Comparison of reconstructing meshes from our completions with NKSR (Huang et al., 2023) using PCA-based estimated normals vs. our estimated normals.

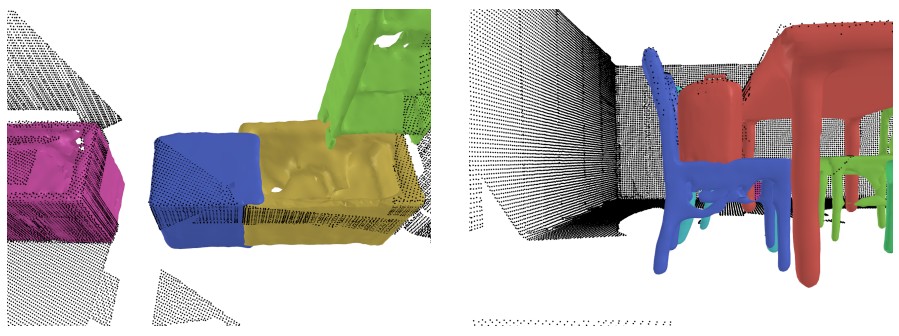

Figure 18: Example failure cases of our scene completion method.

