# OpenReview forum: "Point-based Instance Completion with Scene Constraints"
_ICLR.cc/2025/Conference — ICLR 2025 Poster_

### Official Review · Reviewer_gWgp · 2024-10-16

**Soundness:** 4
**Presentation:** 3
**Contribution:** 3
**Rating:** 8
**Confidence:** 4

**Summary:**

This work leverages joint scene and object information for object point cloud completion in the scene, which is a meaningful and important task. In order to boost completion, the work investigate both object-centric learning and scene occupancy information to guide the completion. In detail, it uses object transformers to generate patch seed in object-centric coordinate with an additional object coordinate, and use scene transformers to represent scene occupancy information. Then, a cross attention is used to combine the information for completion.

The experiments suggest significant improvement using the proposed method.

**Strengths:**

The authors address the point cloud completion task in the scene, which is a fundamental problem in 3D vision. They offer insights that the key for completion is to extract object-centric information in the scene, and combine useful global scanning information such as scene visibility/occupancy.

Accordingly, they provide a network that allow both insights to be implemented.

The idea is clear and the method is solid. Besides, I believe the design shows sufficient technical novelty that reaches the conference standard.

**Weaknesses:**

It may be worth discussing the limitation and quality of point cloud completion using the proposed method under different scenario, e.g., what is the requirement for noise/incompletion ratio for the method to succeed? Can it complete unseen objects?

**Questions:**

Check the weakness part.

---

> ### Author Response · Authors · 2024-11-24
>
> We thank reviewer gWgp for their suggestions and questions. Our responses to the feedback are provided below.
>
>
> ---
> **Subject:  Discussing completion quality under different scenarios (e.g., incompleteness, noise)**
>
> As suggested, we have conducted an experiment where we see how the completion quality of our method varies under different levels of incompleteness in the partial input. In Figure 12a of our appendix, we present a histogram breaking down the object instances in the test set of our dataset by how much of the complete shape is present in the partial object scan. We find that a majority of our partial object scans contain between 30-60% of the complete geometry before being input to our completion model. In Figures 12(b-d), we plot the average completion metrics for each bin in the histogram. We find that our method outperforms the baseline approaches at each completeness level by a large margin, which is consistent with the large gap in performance observed in Tables 1 and 2. However, we do observe that RfD-Net and DIMR do outperform our method for extremely sparse inputs (e.g., when only 0-10% of the object is present in the partial scan). Qualitatively, RfD-Net and DIMR tend to generate a nondesciptive blob whereas our method usually just returns the partial input, leading to them obtaining better completion metrics in those cases, which we think are probably of less interest because of the lack of object information from the observed areas.
>
> In terms of noise, we added  Figure 13 where we visualize some example completions when there are errors present in the instance segmentation predictions produced by Mask3D. In the top four rows, we show examples where Mask3D produced segmentations that missed large or important parts of the partial object instance. The regions missed by Mask3D contain important cues for the true geometry and size of the object, and the lack of this information leads our model to produce a completion different from the ground truth. In the bottom two rows, we show examples where Mask3D incorrectly segments two objects that are side by side as a single object, and our completion model simply followed it and  completed both objects together as if it were one.
>
> ---
> **Subject: Completing unseen objects**
>
> Our method is capable of generalizing to novel objects within the categories we train on, with many of the test scenes in our dataset containing object instances that were never seen during training. We do not evaluate on novel object categories that are not similar to the training categories as we don't believe the method will generalize to those. Instead, in Figure 14 of our appendix, we have additionally shared some examples of how our method generalizes to partial objects from real scans in ScanNet. The objects in ScanNet are novel and their partial scans are considerably less clean than the scans found in our proposed dataset, yet our method is still able to produce plausible completions of the objects.

---

### Official Review · Reviewer_eRJX · 2024-10-31

**Soundness:** 3
**Presentation:** 2
**Contribution:** 3
**Rating:** 6
**Confidence:** 3

**Summary:**

This paper proposed an instance scene completion method for partial point cloud which consists of 3 stages: instance segmentation stage which extract each object from partial point cloud; object completion stage which predicts both complete shape and surface normals for each object; and mesh reconstruction stage which reconstruct scene meshes from point clouds. The focus of this paper is to propose a set of improvement and refinement modules to improve the object completion network. Furthermore, a new dataset for instance scene completion in indoor scenes has been proposed which has better ground truth labels than existing ScanARCW dataset. Experiment on the newly proposed dataset shows that the proposed method achieves SOTA performance on instance scene completion task.

**Strengths:**

This paper is a good practice to build a instance scene completion framework based on the core module of encoder-decoder object completion. There are several design improvements over existing work in the proposed method:
1. Use VI-PointConv to replace PointConv in the partial encoder module in object completion stage.
2. Use object center and offset prediction MLP to generate aligned seed patch in the Seed Generator module in object completion stage.
3. Incorporate the scene constraints into the Seed Generator module through cross-attention
4. Use global attention in upsample layers to improve the Coarse-to-Fine Decoder module in object completion stage.
5. Use surface normal prediction module which enables mesh reconstruction using existing NKSR method
All the above improvements have been verified in ablation study

Experiment shows that the proposed method significantly outperforms existing instance scene completion methods RfD-Net and DIMR (table 1&2)

The paper also proposed a refined dataset 'ScanWCF' which is declared to have 'Watertight and Collision Free' ground-truth, this could benefit future research on instance scene completion task

**Weaknesses:**

There are some concerns on the novelty and effectiveness of the declared main contributions, as the paper said, there are 3 main contributions, the first contribution is 'a novel object-level completion model which is robust to the scale and pose of objects found within scenes', but actually, the whole framework is to deal with instance scene completion, and in instance scene completion, there is no need to constrain the scale and pose of the objects in the scene. the second contribution is 'integrate a sparse set of scene constraints into our model to provide our object completions with scene context', while the corresponding ablation study in table 4 shows limited improvement on UHD and %COL metric. I think the presentation of this paper could be further improved to make the paper stronger

In the ablation study which focus on the object completion task (table 3), the performance improvement of each proposed module is relatively small and not very impressive compared with that on the scene completion task (Table 1&2), furthermore, each module is added separately, what is the performance with all modules included?

**Questions:**

Since existing instance scene completion method jointly train instance segmentation & completion model, while the proposed method uses existing Mask3D method for instance segmentation separately, is there any experiment comparison on the performance of instance segmentation? I may want to see whether the biggest peformance improvement is due to the use of Mask3D?

---

> ### Author Response · Authors · 2024-11-24
>
> We thank reviewer eRJX for their comments and questions. Our responses to the feedback are provided below.
>
> ----
> **Subject: Clarifying contribution about robustness to object pose and scale**
> Our approach extends prior work on object-level completion methods such as SnowflakeNet [1], SeedFormer [2], AnchorFormer [3] (ln. 53 - 63 of our paper). These approaches produce high quality completions but expect the object to be in a canonical coordinate system. Hence our contribution statement is with respect to those approaches where we lifted the requirement of the canonical coordinate system and additionally leveraged scene constraints. Alternatively, prior instance scene completion methods like RfD-Net [4] and DIMR [5] indeed do not require objects to be in canonical coordinates, but these approaches produce much lower quality completions than our approach (since their network design was inferior to the object-level completion methods), as shown in our experiments.
>
> ---
> **Subject: Effectiveness of scene constraints**
> For the scene constraint ablation results in Table 4, we note that we do not expect an improvement in UHD (partial reconstruction quality) because the completion model does not need information about free space and occluded space to reconstruct the already observed portion of the object. However, we do find that including scene constraints significantly improves overall completion quality (CD) by 1.63 (7% relative improvement). While we did not observe much of an improvement in the percent of points in collision (%COL) because most of the points are simply far away from other objects, we do observe a very significant 29% relative improvement (2.67 vs 3.75) in how far these points that are in collision are penetrating into other objects (COL).
>
> ---
> **Subject: Performance improvements in object completion model ablation**
> As mentioned in the response to the first question, the quality of previous instance scene completion methods were significantly inferior to the object-level ones, but object-level ones weren't applicable to the instance scene completion problem before our contributions. From Table 3, we note a 12% relative drop in performance when object-level completion methods are naively used in instance scene completion (20.20 to 22.87). Our approach completely erased this drop and resulted in a significant 12% relative improvement (20.08 vs. 22.87) over the baseline. Note that results in each row contain all the modules in the rows above them. Hence, the final row of Table 3 is the performance of our full completion model. We also note that this ablation was only conducted on the chair category from ShapeNet as training on the 34 ShapeNet categories would be too computationally expensive with the resources we have available. Hence the  numbers presented in Table 3 are not directly comparable to the numbers in Table 2.
>
> ---
> **Subject: Concerns regarding performance improvements being due to better instance segmentation**
> Rather than showing instance segmentation performance, we conducted an experiment evaluating completion quality when the ground truth segmentations are provided to each method (referred to as the "Scene Completion" task in our paper, ln. 449-452). This removes the differences in segmentation quality across methods and isolates the evaluation of completion quality. Results of this experiment in Table 2 show that when all methods are using the same ground truth segmentations, our method significantly outperforms prior work in terms of completion quality.
>
>
> ---
> **References**
> $\left [1 \right]$ Xiang, Peng, et al. "Snowflakenet: Point cloud completion by snowflake point deconvolution with skip-transformer." Proceedings of the IEEE/CVF international conference on computer vision. 2021.
> $\left [2 \right]$ Zhou, Haoran, et al. "Seedformer: Patch seeds based point cloud completion with upsample transformer." European conference on computer vision. Cham: Springer Nature Switzerland, 2022.
> $\left [3 \right]$ Chen, Zhikai, et al. "Anchorformer: Point cloud completion from discriminative nodes." Proceedings of the IEEE/CVF conference on computer vision and pattern recognition. 2023.
> $\left [4 \right]$ Nie, Yinyu, et al. "Rfd-net: Point scene understanding by semantic instance reconstruction." Proceedings of the IEEE/CVF Conference on Computer Vision and Pattern Recognition. 2021.
> $\left [5 \right]$ Tang, Jiaxiang, et al. "Point scene understanding via disentangled instance mesh reconstruction." European conference on computer vision. Cham: Springer Nature Switzerland, 2022.

---

> > ### Comment · Reviewer_eRJX · 2024-11-25
> >
> > Thanks the authors for the detailed explanation of experiment result, my concern on the effectiveness of scene constraint and ablation study has been addressed.  As a result, I tend to increase my rating to 6. I recommend the author to update the manuscript with above explanation on novelty, model ablation setting and scene constraint.

---

> > > ### Author Response · Authors · 2024-11-28
> > >
> > > Thank you once again for your thoughtful comments and questions. We are glad to hear that we were able to address your concerns in regard to experiments. As suggested, we have revised some of the main paper text to reflect some of the discussion we provided as response to your questions and concerns.

---

### Official Review · Reviewer_TJHm · 2024-11-03

**Soundness:** 3
**Presentation:** 3
**Contribution:** 3
**Rating:** 6
**Confidence:** 4

**Summary:**

This paper argue that existing instance scene completion methods may neglect scene constraints when doing instance completion. To address these issues, they propose a point cloud-based instance completion model that utilizes scene constraints through a cross-attention mechanism for improved object completion at any scale or pose. Besides they introduce a new synthetic dataset, ScanWCF, to evaluate instance scene completion in indoor environments. Experiments show that their method outperforms existing techniques in terms of accuracy and plausibility.

**Strengths:**

I think the major contribution of the seed generator converted on top of SeedFormer, that allows the generator to use information from the entire scene to predict the center of the object before producing seed coordinates as offsets of the object center.

Another is the incorporation of a sparse set of scene constraints for seed generator and completion, providing scene context (e.g., other observed surfaces, free space, occluded space).

**Weaknesses:**

I think this is a good paper with sound method and formulation. However, from the technical contribution's perspective, this paper still follows the same "top-down" architecture as in RfDNet, and the performance's boost against baselines comes from the better quality of each individual module. The module this paper is mostly contributing is the object points completion module that completes object pointcloud from partial points given by leveraging the scene context/constraints.

**Questions:**

I am curious that if there are no scene constraints, how would the object completion module perform against the baselines?

---

> ### Author Response · Authors · 2024-11-24
>
> We thank reviewer TJHm for their comments and questions. Our responses to the feedback are provided below.
>
> ---
> **Subject: Framework and Contributions**
> We follow the "segment then complete" paradigm as we believe it is a reasonable framework for solving the instance scene completion task, since it allows methods to use state-of-the-art instance segmentation models which might be trained on much larger data. The major issue with existing works is that they are unable to produce high quality completions due to their poor completion network design. Hence, we focused on improving the object completion module. Aside from the proposed improvements in our object completion model, we have also introduced:
> 1. A way to represent known scene constraints as a sparse set of points, which provide our object completions with context of the scene they reside in.
> 2. A new dataset for the instance scene completion task, which contains aligned partial scans and ground truth meshes of scenes which are collision free. Unlike previous datasets, which either lack alignment or are not collision free, our dataset is more reliable for evaluating completion quality and plausibility.
>
> ---
> **Subject: Performance against baselines without scene constraints**
> Table 4 shows the results of our approach without scene constraints on the scene completion task (same task as the results shown in Table 2). Even without scene constraints, our approach outperforms the baseline methods due to a stronger completion network. However, in Table 4 we demonstrate that scene constraints further improves completion quality (CD) and reduces how badly objects in collision penetrate into each other (COL). Qualitative results were shown in Fig. 7.

---

### Official Review · Reviewer_M7dZ · 2024-11-04

**Soundness:** 3
**Presentation:** 3
**Contribution:** 3
**Rating:** 8
**Confidence:** 4

**Summary:**

This paper presents a scene-level instance completion method based on point clouds. The main contributions are two-fold. First, a more sophisticated seed generator is used to obtain patch seeds. It predicts the offsets to the centers of the objects and leverages scene constraints to generates more reasonable seeds. Second, the authors construct a new dataset for scene instance completion task, which contains watertight and collision-free ground truths. The authors conduct comprehensive experiments and ablation studies to demonstrate the effectiveness of the method.

**Strengths:**

1. The paper is well-written and easy to follow.
2. The illustrations are clear and help understand the paper.
3. The paper has new dataset contributions.
4. The experiment results are promising.

**Weaknesses:**

1. In Line 208, the authors claims that the local attention-based upsampler is worse than the global attention-based method when the objects are not canonical. I cannot see the logic behind. Maybe this is because the proposed method uses VI-PointConv in the encoder? I think the authors should provide more detailed investigation of this.
2. Some important details are missing.
- In Line 203, how does the global shape descriptor generated?
- In Line 206, why are the seeds called the patch seeds? There should be an explanation here.
- In Line 237, how is the tranposed convolution used?
3. How does the proposed method generalize to unknown objects during training?
4. The organization of the paper should be improved. For example, scene-aware object completion (Sec 3.3) is part of the seed generator (Sec 3.2), and describing them separately makes the paper somehow confusing.

**Questions:**

Please address the questions in the weaknesses section.

---

> ### Author Response · Authors · 2024-11-24
>
> We thank reviewer M7dZ for their comments and questions. Our responses to the feedback are provided below.
>
> ---
> **Subject: Local attention vs. global attention-based completion**
> The underlying logic is that one has to see the entirety of the object in order to predict the center. It's difficult for local attention methods to do so with variable object sizes. As mentioned in lines 207 - 210, this was experimentally shown in Table 3, where our "baseline", which uses the local attention-based upsampler from SeedFormer [1], obtains a CD of 20.20 when objects are in canonical coordinates and a CD of 22.87 when they are not. From here, our global attention-based object center prediction results in a 7% relative improvement in completion quality (22.87 to 21.11), and our global attention in upsampling further provides another 5% relative improvement (21.11 to 20.05).
>
> ---
> **Subject: Description of global shape descriptor**
> The global shape descriptor is generated by passing the final local features $F_{p}^{l}$ through a MLP followed by max-pooling. We originally mentioned this in section $B.1$ of our appendix, but have updated the main paper to contain this detail.
>
> ---
> **Subject: Explanation of Patch Seeds**
> "Patch Seeds" is a term coined by SeedFormer [1] and we similarly use this term as our method builds off their architecture. We note that we have already provided a high level description of what Patch Seeds are in lines 157-158 and a formal definition in lines 204-206 of our main paper. In particular, Patch Seeds are a set of coordinates $S \in \mathbb{R}^{M_{seed} \times 3}$ and corresponding features $F_{seed} \in \mathbb{R}^{M_{seed} \times C_{seed}}$, which represent a coarse encoding of the complete shape.
>
> ---
> **Subject: Purpose of transposed convolution**
> We note that the purpose of the transposed convolution is described in lines 236 - 239. In particular, we want to increase the number of points present in our Patch Seeds compared to our downsampled partial points and features that they are generated from. The transposed convolution is used to upsample or "split" the existing set of features we have. We specifically use a transposed 1D convolution with stride 2 and kernel size 2, lifting the features from dimensions $N \times C$ to $2N \times C$ (i.e., we double the number of points).
>
> ---
> **Subject: Generalizing to unseen objects**
> We assume the reviewer meant during inference time as it is not possible for the model to generalize to unknown objects during training. Our method is capable of generalizing to novel objects within the categories we train on, with many of the test scenes in our dataset containing object instances that were never seen during training. We do not evaluate on novel object categories that are not similar to the training categories as we don't believe the method will generalize to those. Instead, in Figure 14 of our appendix, we have additionally shared some examples of how our method generalizes to partial objects from real scans in ScanNet. The objects in ScanNet are novel and their partial scans are considerably less clean than the scans we trained on from our dataset, yet our method is still able to produce plausible completions of the objects.
>
> ---
> **References**
> $\left [1 \right]$ Zhou, Haoran, et al. "Seedformer: Patch seeds based point cloud completion with upsample transformer." European conference on computer vision. Cham: Springer Nature Switzerland, 2022.

---

> ### Comment · Reviewer_M7dZ · 2024-12-03
>
> Thanks the authors for the clear clarification. Most of my concerns have been addressed, and I would raise my overall score from 6 to 8.

---

### Author Response · Authors · 2024-11-24

We thank the reviewers for taking the time to read and review our work. We have tried our best to answer and address any questions and clarifications in each individual reviewer response.

Additionally, we would like to point out that the One-Sided Chamfer Distance numbers reported in Table 2 of our paper were originally incorrect. We found an error in our computation, where we had accidentally computed the One-Sided Chamfer Distance metric between the ground truth shape and predicted completion instead of the partial input and predicted completion. We have updated Table 2 in the paper to reflect the correct numbers for the One-Sided Chamfer Distance metric. We have also provided a comparison below between the old and new numbers. With this correction, we observe that our method obtains an even larger improvement in partial reconstruction performance over the baseline methods than was previously observed in our original results.

|                        | One-Sided CD  (Old) | One-Sided CD  (New) |
|------------------------|:-------------------:|:-------------------:|
| RfD-Net                |        32.72        |        38.39        |
| DIMR                   |        38.28        |        44.91        |
| Ours (no pre-taining)  |        22.83        |        12.50        |
| Ours (w/ pre-training) |        22.38        |      **12.10**      |

---

### Meta-Review · Area_Chair_emi3 · 2024-12-14

**Metareview:**

The authors present a method for scene completion from given scans and scene constraints. The paper was well-received with positive scores by all reviewers. Mentioned positive aspects include the overall paradigm design, the way to introduce scene constraints and the clear improvements over previous works.
I see no reason to object and follow with an accept recommendation.

**Additional Comments On Reviewer Discussion:**

Initially, the reviewers pointed our some unclear details and were requesting additional ablation studies. The authors addressed these concerns in their answers. The reviewers were satisfied and increased or kept their score.

---

### Decision · Program_Chairs · 2025-01-22

Accept (Poster)